

# Database of nitrification and nitrifiers in the global ocean
Weiyi Tang[1], Bess B. Ward[1], Michael Beman[2], Laura Bristow[3], Darren Clark[4], Sarah Fawcett[5],
Claudia Frey[6], Francois Fripiat[7], Gerhard J. Herndl[8], Mhlangabezi Mdutyana[5], Fabien Paulot[9],
Xuefeng Peng[10], Alyson E. Santoro[11], Takuhei Shiozaki[12], Eva Sintes[13], Charles Stock[9], Xin Sun[14],
Xianhui S. Wan[1], Min N. Xu[15], Yao Zhang[16]
Affiliations:
1. Department of Geosciences, Princeton University, Princeton, NJ 08544, USA
2. Life and Environmental Sciences, University of California, Merced, Merced, CA, USA
3. Department of Marine Sciences, University of Gothenburg, Gothenburg, Sweden
4. Somserset Scientific Services, Westpark 26, Chelston, Wellington, Somerset TA21 9AD, UK
5. Department of Oceanography, University of Cape Town, Rondebosch 7701, South Africa
6. Department of Environmental Science, University of Basel, Basel, Switzerland
7. Department of Geosciences, Environment and Society, Université Libre de Bruxelles,
Brussels, Belgium
8. Department of Functional and Evolutionary Ecology, University of Vienna, Vienna, Austria
9. Geophysical Fluid Dynamics Laboratory, National Oceanic and Atmospheric Administration,
Princeton, NJ, USA
10. School of Earth, Ocean and Environment, University of South Carolina, Columbia, SC
29208, USA
11. Department of Ecology, Evolution and Marine Biology, University of California, Santa
Barbara, Santa Barbara, CA, USA
12. Atmosphere and Ocean Research Institute, The University of Tokyo, Chiba, Japan
13. Instituto Español de Oceanografía-CSIC, Centro Oceanográfico de Baleares, Palma de
Mallorca, Spain
14. Department of Global Ecology, Carnegie Institution for Science, Stanford, CA, USA
15. State Key Laboratory of Marine Resource Utilization in South China Sea, Hainan University,
Haikou 570228, China
16. State Key Laboratory of Marine Environmental Sciences, Xiamen University, Xiamen
361101, China



33 *Correspondence to*: Weiyi Tang (weiyit@princeton.edu)



## Abstract

As a key biogeochemical pathway in the marine nitrogen cycle, nitrification (ammonia oxidation and nitrite oxidation) converts the most reduced form of nitrogen – ammonium/ammonia ($NH_4^+$/ $NH_3$) into the oxidized species nitrite ($NO_2^-$) and nitrate ($NO_3^-$). In the ocean, these processes are mainly performed by ammonia-oxidizing archaea (AOA) and bacteria (AOB), and nitrite-oxidizing bacteria (NOB). By transforming nitrogen speciation and providing substrates for nitrogen removal, nitrification affects microbial community structure, marine productivity (including chemoautotrophic carbon fixation) and the production of a powerful greenhouse gas, nitrous oxide ($N_2O$). Nitrification is hypothesized to be regulated by temperature, oxygen, light, substrate concentration, substrate flux, pH, and other environmental factors. Although the number of field observations from various oceanic regions has increased considerably over the last few decades, a global synthesis is lacking, and understanding how environmental factors control nitrification remains elusive. Therefore, we have compiled a database of nitrification rates and nitrifier abundance in the global ocean from published literature and unpublished datasets. This database includes 2393 and 1006 measurements of ammonia oxidation and nitrite oxidation rates, and 2187 and 631 quantifications of ammonia oxidizers and nitrite oxidizers, respectively. This community effort confirms and enhances our understanding of the spatial distribution of nitrification and nitrifiers, and their corresponding drivers such as the important role of substrate concentration in controlling nitrification rates and nitrifier abundance. Some conundrums are also revealed including the inconsistent observations of light limitation and high rates of nitrite oxidation reported from anoxic waters. This database can be used to constrain the distribution of marine nitrification, to evaluate and improve biogeochemical models of nitrification, and to quantify the impact of nitrification on ecosystem functions like marine productivity and $N_2O$ production. This database additionally sets a baseline for comparison with future observations and guides future exploration (e.g., measurements in the poorly sampled regions such as the Indian Ocean; method comparison/standardization). The database is publicly available at Zenodo repository: https://doi.org/10.5281/zenodo.7942922 (Tang et al., 2023).



## Introduction

Nitrification (ammonia oxidation and nitrite oxidation) converts the most reduced form of nitrogen (N) – ammonium/ammonia ($NH_4^+$/$NH_3$) into the oxidized compounds nitrite ($NO_2^-$) and nitrate ($NO_3^-$). Ammonia oxidation is conducted by ammonia oxidizing archaea (AOA) and bacteria (AOB) with AOA dominating in most marine environments (Francis et al., 2005; Wuchter et al., 2006). Marine AOA are often separated into a few major ecotype groups including water column group A, water column group B and *Nitrosopumilus*-like (Beman et al., 2008; Tolar et al., 2020), with a diverse goup of AOA remaining to be characterized (Alves et al., 2018). Marine nitrite oxidation is carried out by nitrite-oxidizing bacteria (NOB) such as *Nitrospina*, *Nitrospira*, *Nitrococcus* and *Nitrobacter*, with *Nitrospina* as the dominant group (Mincer et al., 2007; Pachiadaki et al., 2017). Complete ammonia-oxidizing (comammox) bacteria within the bacterial genus *Nitrospira* have been identified in freshwater, terrestrial, and coastal environments but not yet been found in the open ocean (Daims et al., 2015; Van Kessel et al., 2015; Xia et al., 2018).

Nitrification and nitrifiers are thought to be regulated by light/solar radiation, oxygen, temperature, substrate concentration, pH, and other environmental factors (Ward, 2008), many of which are experiencing dramatic changes in the ocean. For example, light is generally found to inhibit nitrifiers' growth and nitrification rate (Olson. 1981b; Merbt et al., 2012; Xu et al., 2019). In addition, ocean acidification decreases ammonia oxidation rates (Beman et al., 2011; Breider et al., 2019) partly due to the decreased availability at lower pH of $NH_3$, which is the actual substrate for ammonia oxidation (Suzuki et al., 1974). In contrast, ocean warming shifts the $NH_4^+$/$NH_3$ equilibrium towards $NH_3$ by decreasing the *pKa* (Emerson et al., 1975) and is observed to enhance enzyme activity (Zheng et al., 2017; Zheng et al., 2020), further complicating the effect of climate change on nitrification.

Although nitrification does not directly change the absolute inventory of bioavailable N, it can control the relative availability of substrates ($NH_4^+$, $NO_2^-$ and $NO_3^-$) for phytoplankton growth. Since prokaryotic phytoplankton preferentially assimilate $NH_4^+$ while eukaryotic phytoplankton are better able to exploit $NO_3^-$ in the sunlit surface ocean (Berthelot et al., 2018; Fawcett et al., 2011), variations in the relative supply of $NH_4^+$ versus $NO_3^-$ can influence phytoplankton community composition and ecosystem functionalities. Because the uptakes of $NH_4^+$ and $NO_3^-$ are



often used to differentiate regenerated and new production (Eppley and Peterson. 1979),
production of $NO_3^-$ by nitrification in the surface ocean may bias the estimate of new production
(Yool et al., 2007). $NO_2^-$ and $NO_3^-$ are also involved in denitrification and anammox, which remove
bioavailable N from the ocean. Thus, nitrification can indirectly affect the size of the bioavailable
N pool, marine productivity and ultimately the atmospheric $CO_2$ concentration (Falkowski, 1997).
As a chemoautotrophic process, nitrification in the ocean water column is estimated to supply
~0.13-1.4 Pg C $yr^{-1}$ of organic matter, which is critical to support the heterotrophic microbial
community/metabolism in the dark ocean (Bayer et al., 2022; Middelburg, 2011; Pachiadaki et al.,
2017; Zhang et al., 2020). Nitrification could also contribute to the oxygen consumption and the
development of hypoxia or anoxia (Hsiao et al., 2014; Beman et al., 2021). In addition, nitrification
is the major global ocean source of $N_2O$, a potent greenhouse gas and dominant ozone-depleting
agent, thus connecting the marine N cycle directly to the Earth's climate system (Freing et al.,
2012; Ji et al., 2018).

Considering the important role of nitrification and nitrifiers in marine N and C cycles and Earth's
climate, a better understanding of its distribution and regulating factors is highly desirable.
Historical observations of nitrification and nitrifiers cover a wide range of environmental gradients
and biogeography in the ocean, ranging from cross-Atlantic (e.g., Clark et al., 2008; Clark et al.,
2022), western Pacific (e.g., Wan et al., 2021; Wan et al., 2018), polar oceans (e.g., Shiozaki et
al., 2019; Mdutyana et al., 2020) to oxygen minimum zones (e.g., Peng et al., 2015; Santoro et al.,
2021). This study aims to introduce the newly constructed database of nitrification and nitrifiers
in the marine water column and to guide future research efforts in field observations and model
development of nitrification. This new global synthesis significantly expands upon what was
possible with earlier more limited datasets (Yool et al. 2007; Ward. 2008). Additional reviews on
marine nitrification and nitrifiers can be found elsewhere (Schleper and Nicol, 2010; Daims et al.,
2016; Ward, 2011b).



## Methods

### Data sources and compilation

Nitrification rates including ammonia oxidation and nitrite oxidation, and abundances of ammonia oxidizers and nitrite oxidizers were extracted directly from the literature published between 1984 and 2022 when the data were presented in tables or supplementary materials from publications; otherwise, data were provided by the coauthors. Some previously unpublished data were also included in the database. Table 1 and Table 2 summarize the origin, methods and locations of nitrification rate and nitrifier abundance measurements, sorted in alphabetical order by lead author. The metadata format contains geographical sampling information (date, latitude, longitude, and depth) and concurrent measurements of environmental conditions such as light intensity, temperature, salinity, water density, N concentration ($NH_4^+$, $NO_2^-$ and $NO_3^-$), pH and oxygen concentration if available. In total, there are 2393, 1006, 2187, and 631 measurements of ammonia oxidation rate, nitrite oxidation rate, ammonia oxidizer abundance and nitrite oxidizer abundance, respectively. However, not all measurements of nitrification rates or nitrifier abundance are accompanied by all the environmental factors because such factors were often not reported in the literature or recorded during the measurements/sample collections. Rates, nitrifier abundances and environmental parameters below the methodological detection limits are noted as BDL. NM represents parameters that were not measured. Empty/NA means that data are not available or reported. The database is deposited into Zenodo repository following the Findable, Accessible, Interoperable and Reusable (FAIR) principles for data management (Wilkinson et al., 2016). We encourage authors and readers to contact us to report an update to or an error in the database.

Table 1. Summary of the number of observations for nitrification rates in alphabetical order of the lead author. The method (e.g., substrate tracer addition vs product dilution) and sampling regions are listed. Methods used for data collection are described in the next section.

| References | | | Nitrification | | | Sampling regions |
|---|---|---|---|---|---|---|
| | Ammonia oxidation | Method | Analyte | Nitrite oxidation | Method | |
| Baer et al., 2017 | 6 | $^{15}NH_4^+$ tracer addition | $NO_2^-+$ $NO_3^-$ | | | Western Coastal Arctic |





| | | | | | | |
|---|---|---|---|---|---|---|
| Beman et al., 2012 | 68 | $^{15}NH_4^+$ tracer addition | $NO_2^- + NO_3^-$ | 64 | $^{15}NO_2^-$ tracer addition | Eastern Tropical North Pacific |
| Beman et al., 2021 | 78 | $^{15}NH_4^+$ tracer addition | $NO_2^- + NO_3^-$ | 79 | $^{15}NO_2^-$ tracer addition | Eastern Tropical North Pacific |
| Bianchi et al., 1997 | 21 | $H^{14}CO_3^-$ tracer addition | Particulate organic carbon | 21 | $H^{14}CO_3^-$ tracer addition | Southern Ocean |
| Breider et al., 2019 | 10 | $^{15}NH_4^+$ tracer addition | $NO_2^- + NO_3^-$ | | | Western North Pacific |
| Bristow et al., 2015 | 9 | $^{15}NH_4^+$ tracer addition | $NO_2^-$ | 9 | $^{15}NO_2^-$ tracer addition | Gulf of Mexico |
| Cavagna et al., 2015 | | | | 39 | $^{15}NO_3^-$ tracer dilution | Southern Ocean |
| Clark et al., 2008 | 32 | $^{15}NO_2^-$ tracer dilution | $NO_2^-$ | 32 | $^{15}NO_3^-$ tracer dilution | Atlantic |
| Clark et al., 2011 | 13 | $^{15}NO_2^-$ tracer dilution | $NO_2^-$ | 13 | $^{15}NO_3^-$ tracer dilution | Eastern North Atlantic (offshore of the Iberian Peninsula) |
| Clark et al., 2014 | 10 | $^{15}NO_2^-$ tracer dilution | $NO_2^-$ | 10 | $^{15}NO_3^-$ tracer dilution | Northwest European shelf sea |
| Clark et al., 2016 | 21 | $^{15}NO_2^-$ tracer dilution | $NO_2^-$ | 42 | $^{15}NO_3^-$ tracer dilution | Mauritanian upwelling system |
| Clark et al., 2022 | 88 | $^{15}NO_2^-$ tracer dilution | $NO_2^-$ | | | Atlantic |
| Clark et al., unpublished 1 | 18 | $^{15}NO_2^-$ tracer dilution | $NO_2^-$ | 18 | $^{15}NO_3^-$ tracer dilution | Eastern North Atlantic |
| Clark et al., unpublished 2 | 18 | $^{15}NO_2^-$ tracer dilution | $NO_2^-$ | 18 | $^{15}NO_3^-$ tracer dilution | Eastern North Atlantic |
| Clark et al., unpublished 3 | 21 | $^{15}NO_2^-$ tracer dilution | $NO_2^-$ | 21 | $^{15}NO_3^-$ tracer dilution | Eastern North Atlantic |
| Clark et al., unpublished 4 | 11 | $^{15}NO_2^-$ tracer dilution | $NO_2^-$ | 11 | $^{15}NO_3^-$ tracer dilution | Subpolar North Atlantic and Arctic |





| Damashek et al., 2018 | 15 | $^{15}NH_4^+$ tracer addition | $NO_2^-+NO_3^-$ | | | South Atlantic Bight |
|---|---|---|---|---|---|---|
| Diaz and Raimbault, 2000 | 20 | $^{15}NH_4^+$ tracer addition | $NO_2^-+NO_3^-$ | | | Gulf of Lions in the Mediterranean Sea |
| Dore and Karl, 1996 | 11 | $NO_2^-+NO_3^-$ concentration change over time; $H^{14}CO_3^-$ tracer addition | $NO_2^-+NO_3^-$, particulate organic carbon | 6 | $NO_3^-$ concentration change over time | Station ALOHA in the North Pacific |
| Fernández et al., 2009 | 15 | $^{15}NH_4^+$ tracer addition | $NO_2^-+NO_3^-$ | | | Peru upwelling system |
| Flynn et al., 2021 | | | | 104 | $^{15}NO_2^-$ tracer addition | Weddell Sea |
| Frey et al., 2020 | 21 | $^{15}NH_4^+$ tracer addition | $NO_2^-$ | | | Eastern Tropical South Pacific |
| Frey et al., 2022 | 30 | $^{15}NH_4^+$ tracer addition | $NO_2^-$ | | | Eastern Tropical North Pacific |
| Ganesh et al., 2015 | 5 | $^{15}NH_4^+$ tracer addition | $NO_2^-$ | 5 | $^{15}NO_2^-$ tracer addition | Eastern Tropical North Pacific oxygen minimum zone |
| Kalvelage et al., 2011 | 6 | $^{15}NH_4^+$ tracer addition | $NO_2^-$ | | | Namibian oxygen minimum zone |
| Kalvelage et al., 2013 | 108 | $^{15}NH_4^+$ tracer addition | $NO_2^-$ | 110 | $^{15}NO_2^-$ tracer addition | Eastern Tropical South Pacific oxygen minimum zone |
| Kitzinger et al., 2020 | 9 | $^{15}NH_4^+$ tracer addition | $NO_2^-$ | 9 | $^{15}NO_2^-$ tracer addition | Gulf of Mexico |
| Lam et al., 2009 | 14 | $^{15}NH_4^+$ tracer addition | $NO_2^-$ | | | Eastern Tropical South Pacific |
| Laperriere et al., 2020 | 59 | $^{15}NH_4^+$ tracer addition | $NO_2^-+NO_3^-$ | | | Southern California Bight |



| Liu et al., 2018 | 86 | $^{15}NH_4^+$ tracer addition | $NO_2^-$+$NO_3^-$ | | | South Atlantic Bight |
|---|---|---|---|---|---|---|
| Liu et al., 2022 | 10 | $^{15}NH_4^+$ tracer addition | $NO_2^-$+$NO_3^-$ | | | South China Sea |
| Mccarthy et al., 1999 | 8 | $^{15}NH_4^+$ tracer addition | $NO_2^-$+$NO_3^-$ | | | Arabian Sea |
| Mdutyana et al., 2020 | 59 | $^{15}NH_4^+$ tracer addition | $NO_2^-$ | 38 | $^{15}NO_2^-$ tracer addition | Southern Ocean |
| Mdutyana et al., 2022a | 24 | $^{15}NH_4^+$ tracer addition | $NO_2^-$ | | | Southern Ocean |
| Mdutyana et al., 2022b | | | | 24 | $^{15}NO_2^-$ tracer addition | Southern Ocean |
| Newell et al., 2013 | 8 | $^{15}NH_4^+$ tracer addition | $NO_2^-$ | | | Sargasso Sea (western North Pacific) |
| Peng et al., 2015 | 30 | $^{15}NH_4^+$ tracer addition | $NO_2^-$, $NO_2^-$+$NO_3^-$ | 30 | $^{15}NO_2^-$ tracer addition | Eastern Tropical North Pacific |
| Peng et al., 2016 | 47 | $^{15}NH_4^+$ tracer addition | $NO_2^-$ | 47 | $^{15}NO_2^-$ tracer addition | Eastern Tropical South Pacific |
| Peng et al., 2018 | 28 | $^{15}NH_4^+$ tracer addition | $NO_2^-$ | 28 | $^{15}NO_2^-$ tracer addition | Subarctic North Atlantic |
| Raes et al., 2020 | 39 | $^{15}NH_4^+$ tracer addition | $NO_2^-$+$NO_3^-$ | | | South Pacific |
| Raimbault et al., 1999 | 41 | $^{15}NH_4^+$ tracer addition | $NO_2^-$+$NO_3^-$ | | | Equatorial Pacific |
| Santoro et al., 2010 | 11 | $^{15}NH_4^+$ tracer addition | $NO_2^-$+$NO_3^-$ | | | Central California Current |
| Santoro et al., 2013 | 10 | $^{15}NH_4^+$ tracer addition | $NO_2^-$, $NO_2^-$+$NO_3^-$ | | | Central California Current |
| Santoro et al., 2017 | 12 | $^{15}NH_4^+$ tracer addition | $NO_2^-$+$NO_3^-$ | | | Equatorial Pacific |
| Santoro et al., 2021 | 57 | $^{15}NH_4^+$ tracer addition | $NO_2^-$+$NO_3^-$ | 57 | $^{15}NO_2^-$ tracer addition | Eastern Tropical South Pacific |



| Sinyanya et al., unpublished | | | | 31 | $^{15}NO_2^-$ tracer addition | Southwest Indian Ocean |
|---|---|---|---|---|---|---|
| Shiozaki et al., 2016 | 87 | $^{15}NH_4^+$ tracer addition | $NO_2^-$+$NO_3^-$ | | | Equatorial Pacific to the Arctic Ocean |
| Shiozaki et al., 2019 | 56 | $^{15}NH_4^+$ tracer addition | $NO_2^-$+$NO_3^-$ | | | Arctic Ocean |
| Shiozaki et al., 2021 | 28 | $^{15}NH_4^+$ tracer addition | $NO_2^-$+$NO_3^-$ | | | Arctic Ocean |
| Smith et al., 2022 | 11 | $^{15}NH_4^+$ tracer addition | $NO_2^-$ | | | Southern Ocean |
| Sun et al., 2017 | | | | 9 | $^{15}NO_2^-$ tracer addition | Eastern Tropical North Pacific |
| Sutka et al., 2004 | 20 | $^{15}NH_4^+$ tracer addition | $NO_2^-$+$NO_3^-$ | | | North Pacific Subtropical Gyre to Eastern Tropical North Pacific |
| Tolar et al., 2016 | 73 | $^{15}NH_4^+$ tracer addition | $NO_2^-$+$NO_3^-$ | | | Antarctic coast |
| Tolar et al., 2017 | 38 | $^{15}NH_4^+$ tracer addition | $NO_2^-$+$NO_3^-$ | | | Georgia coast, South Atlantic Bight, Gulf of Alaska, Antarctic coast |
| Tolar et al., 2020 | 297 | $^{15}NH_4^+$ tracer addition | $NO_2^-$+$NO_3^-$ | | | Monterey Bay |
| Wallschuss et al., 2022 | 40 | $^{15}NH_4^+$ tracer addition | $NO_2^-$ | 40 | $^{15}NO_2^-$ tracer addition | Southeastern Atlantic |
| Wan et al., 2018 | 90 | $^{15}NH_4^+$ tracer addition | $NO_2^-$+$NO_3^-$ | | | South China Sea and Northwest Pacific |
| Wan et al., 2021 | 17 | $^{15}NH_4^+$ tracer addition | $NO_2^-$ | 17 | $^{15}NO_2^-$ tracer addition | North Pacific |
| Wan et al., 2022 | 85 | $^{15}NH_4^+$ tracer addition | $NO_2^-$ | | | North Pacific |





| Ward et al., 1984 | 16 | $^{15}NH_4^+$ tracer addition | $NO_2^-$ | | | Coastal waters off Washington |
|---|---|---|---|---|---|---|
| Ward, 1987 | 24 | $^{15}NH_4^+$ tracer addition | $NO_2^-$ | | $^{15}NO_2^-$ tracer addition | Southern California Bight |
| Ward and Zafiriou, 1988 | 42 | $^{15}NH_4^+$ tracer addition | $NO_2^-$ | | | Eastern Tropical North Pacific |
| Ward et al., 1989 | 47 | $^{15}NH_4^+$ tracer addition | $NO_2^-$ | 47 | $^{15}NO_2^-$ tracer addition | Eastern Tropical South Pacific |
| Ward, 2005 | 110 | $^{15}NH_4^+$ tracer addition | $NO_2^-$ | | | Monterey Bay |
| Xu et al., 2018 | 78 | $^{15}NH_4^+$ tracer addition | $NO_2^-$ | | | South China Sea |
| Zhang et al., 2020 | 27 | $^{15}NH_4^+$ tracer addition | $NO_2^-$ | 27 | $^{15}NO_2^-$ tracer addition | South China Sea and Western Pacific |
| Total number of observations | 2393 | | | 1006 | | |


Table 2. Summary of the number of observations for nitrifier abundance from qPCR assays in
alphabetical order of the lead authors. The top row indicates the gene quantified for each group
(see text for further details). The primers used for individual studies are identified in the database.
AOA: ammonia-oxidizing archaea; AOB: ammonia-oxidizing bacteria; NOB: nitrite-oxidizing
bacteria.

| References | *amoA*-based | | *nxr*-based | *16S rRNA*-based | | | Sampling regions |
|---|---|---|---|---|---|---|---|
| | AOA | AOB | NOB | *Thaumarchaeota* | *Nitrospira* | *Nitrospina* | |
| Agogue et al., 2008 | 55 | 55 | | 55 | | | North Atlantic |
| Beman et al., 2012 | 64 | 64 | | 64 | | | Eastern Tropical North Pacific |
| Beman et al., 2013 | | | | | | 63 | Eastern Tropical North Pacific |
| Bristow et al., 2016b | 27 | | 27 | | | | Bay of Bengal oxygen minimum zone |



| | | | | | | |
|---|---|---|---|---|---|---|
| Damashek et al., 2018. | 34 | | 34 | | 34 | South Atlantic Bight |
| Frey et al., 2020 | 21 | | | | | South Pacific oxygen minimum zone |
| Frey et al., 2022 | 30 | | | | | North Pacific oxygen minimum zone |
| Horak et al., 2018 | 6 | 6 | | | | North Pacific Ocean |
| Kalvelage et al., 2013. | 143 | 89 | | | | South Pacific oxygen minimum zone |
| Liu et al., 2018. | 385 | 385 | 385 | | 385 | South Atlantic Bight |
| Peng et al., 2013 | 23 | | | | | Eastern Tropical North Pacific |
| Peng et a., 2015 | 19 | 19 | | | | Eastern Tropical South Pacific |
| Peng et a., 2016 | 19 | 19 | | | | Subarctic North Atlantic |
| Santoro et al., 2010 | 17 | 17 | 17 | | 17 | Central California Current |
| Santoro et al., 2013 | 10 | 10 | | | | Central California Current |
| Santoro et al., 2017 | 148 | | | | | Equatorial Pacific |
| Santoro et al., 2021 | 78 | 24 | 78 | | 78 | Eastern Tropical South Pacific |
| Shiozaki et al., 2016 | 87 | 87 | | | | North Pacific |
| Shiozaki et al., 2019 | 56 | 56 | | | | Arctic Ocean |
| Shiozaki et al., 2021 | 28 | 28 | | | | Arctic Ocean |
| Sintes et al., 2013 | 115 | | 115 | | | Tropical Atlantic and coastal Arctic |



| | | | | | | |
|---|---|---|---|---|---|---|
| Sintes et al., 2016 | 364 | | | 364 | | | Atlantic Ocean |
| Tolar et al., 2016 | 73 | 73 | | | | | Antarctic coast |
| Tolar et al., 2017 | 38 | | | 38 | | | Georgia coast, South Atlantic Bight, Gulf of Alaska, Antarctic coast |
| Tolar et al., 2020 | 297 | | | | | | Monterey Bay |
| Wuchter et al., 2006 | 20 | 20 | | 20 | | | Atlantic Ocean |
| Zakem et al., 2018 | 31 | | | | | | North Pacific |
| Zhang et al., 2020 | 54 | 54 | | 54 | 54 | 54 | South China Sea and Western Pacific |
| Total points | 2187 | 951 | 27 | 1169 | 54 | 631 | |


We applied Chauvenet's criterion for quality control to flag outliers in nitrification rates and
nitrifier abundance (Glover et al., 2011). Chauvenet's criterion is commonly applied to normally
distributed datasets to identify outliers whose deviations from the mean have a probability of less
than $1/(2n)$, where $n$ is the number of data points (Buitenhuis et al., 2013). We applied the criterion
acknowledging the fact that the data were collected at different environmental conditions. After
removing measurements of zero and below detection limit, nitrification rates and nitrifier
abundances were log10 transformed before further analysis. Nitrification rates and nitrifier
abundances reported at 0 or below detection limit are noted separately in the database and
following analysis. Although we did not find outliers for ammonia oxidation and nitrite oxidation
rates, there are some extreme values worth noting. For example, an extremely high ammonia
oxidation rate of 4900 nmol $L^{-1}$ $d^{-1}$ was observed in the Peruvian oxygen minimum zone (Lam et
al., 2009). Low but detectable rates below 0.01 nmol $L^{-1}$ $d^{-1}$ were observed in the Eastern Tropical
North Pacific oxygen minimum zone (Frey et al., 2022), South Atlantic Bight (Liu et al., 2018)
and western Pacific (Xu et al., 2018). Some outliers were identified by Chauvenet's criterion for
ammonia oxidizers. For instance, an abnormally high abundance of the bacterial *amoA* gene ($10^8$
copies $L^{-1}$) was observed in the South Pacific oxygen minimum zone (Kalvelage et al., 2013),
which was removed from the following analysis. A low abundance of 16S rRNA of
*Thaumarchaeota* (25 copies $L^{-1}$) was found in the surface water of the western Pacific (Zhang et





al., 2020). In addition, the low-ammonia concentration AOA ecotype (or water column group B
AOA) at 2 copies $L^{-1}$ was reported in the Arctic Ocean (Sintes et al., 2013). Measurements of
nitrification rate and nitrifier abundance of 0 or below detection limit were not included in the
analysis of outlier identification. For example, AOA abundance at 0 or below detection limit
(varies among studies) has been reported in surface waters of South Atlantic Bight (Damashek et
al., 2018), equatorial Pacific (Santoro et al., 2017) and North Pacific (Shiozaki et al., 2016).

**Methods for measuring ammonia oxidation and nitrite oxidation rates**
Ammonia oxidation rate is commonly measured by comparing the change in nitrite ($NO_2^-$) and
nitrate ($NO_3^-$) concentration in controls versus an experimental treatment containing a nitrification
inhibitor (e.g., Dore and Karl, 1996), by tracking the oxidation of $^{15}NH_4^+$ into the $NO_2^-$ and $NO_3^-$
pool (Olson, 1981a), or by the dilution of $^{15}NO_2^-$ (Clark et al., 2007). Similarly, nitrite oxidation
rate can be measured by the change in $NO_3^-$ concentration, by tracking the oxidation of $^{15}NO_2^-$ into
the $NO_3^-$ pool, or by the dilution of $^{15}NO_3^-$ (Ward et al., 1989). In addition, nitrification has also
been estimated from the incorporation of $^{14}C$ tracer due to the chemoautotrophic metabolism of
nitrifiers (Bianchi et al., 1997). There is a large uncertainty, however, in the conversion factor from
carbon fixation to nitrification (Bayer et al., 2022). A more detailed description of methods for
measuring nitrification can be found in Ward, 2011a. The spatial distribution of different methods
used to measure nitrification and the frequency distribution of measured rates by different methods
are shown in Figure 1. Rates measured with the substrate tracer addition method ($^{15}NH_4^+$ and
$^{15}NO_2^-$) outnumbered other methods globally but the product dilution method ($^{15}NO_2^-$ and $^{15}NO_3^-$)
dominated in the Atlantic Ocean. The ammonia oxidation rates measured by different methods
have similar median values. However, the median nitrite oxidation rate measured by the $^{15}NO_3^-$
dilution method is significantly higher than the rate measured by the $^{15}NO_2^-$ addition method (200.3
vs 7.4 nmol N $L^{-1}$ $d^{-1}$). These comparisons, however, are between samples aggregated from
measurements taken at different sites. It is thus unclear whether the differences arise from
differences in the measurement approaches (e.g., in sensitivity) or in the sites where measurements
were made. A direct methods comparison is recommended for future exploration.

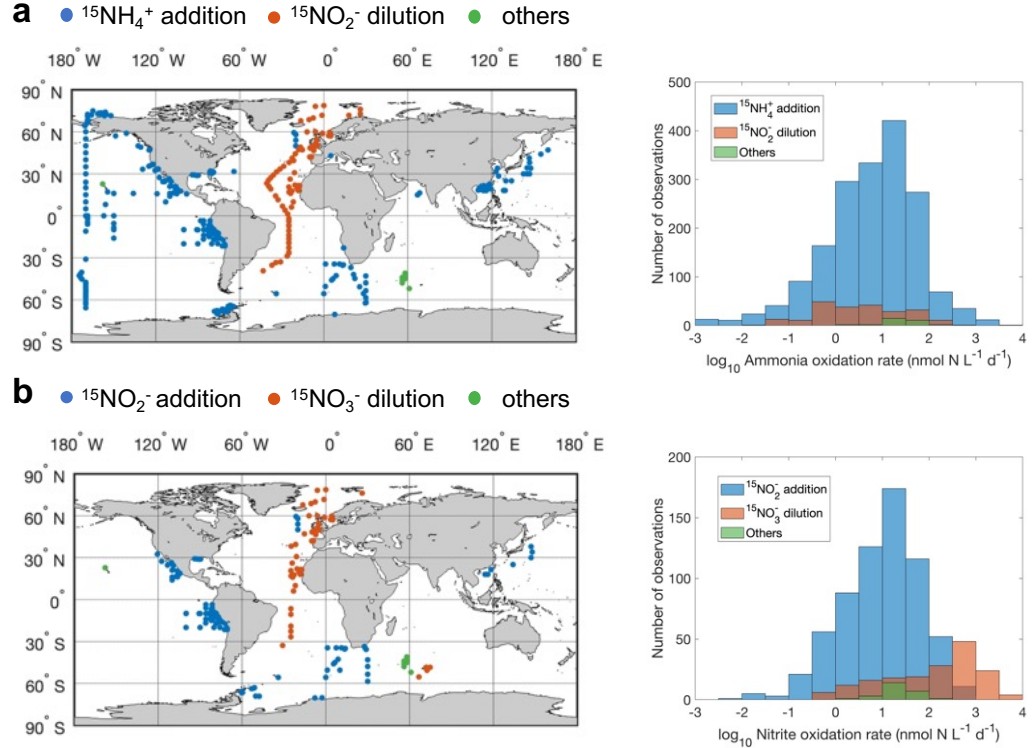

Figure 1. Distribution of different methods used to measure ammonia oxidation (a) and nitrite oxidation (b). Others include $^{14}C$ incorporation and concentration change methods.

Incubations to measure nitrification rates have been conducted in polycarbonate and glass bottles, exetainers and plastic bags. Seawater is directly transferred from the Niskin bottle into the incubation containers to minimize temperature, oxygen and other perturbations. These incubation containers are usually kept in an incubator with light filters to mimic the ambient temperature and light conditions. After incubating for 3 hours to over 24 hours depending on the estimated magnitude of nitrification rates, the incubation is terminated by filtering via GF/F or 0.22 μm filters (e.g., Baer et al., 2017; Wan et al., 2019). The filtrate is then frozen at -20ºC or -80ºC until further analysis on land. The incubation has also been terminated by subsampling and freezing without filtration (e.g., Damashek et al., 2018). Alternatively, the incubation is preserved by adding mercury chloride or zinc chloride (Kalvelage et al., 2013; Frey et al., 2020). This method allows gas measurements like $N_2O$ and $N_2$ production before nitrification analysis.





Various approaches have been developed to measure the N isotopes of $NO_2^-$ and $NO_3^-$. For
example, 1) dissolved $NO_2^-$ is extracted by formation of an azo dye. The resulting dye is filtered
onto precombusted GF/F or GF/C filters and its $^{15}N$:$^{14}N$ ratio is analyzed by elemental analyzer
isotope ratio mass spectrometry (Ward et al., 1982; Olson, 1981a). $NO_3^-$ can be reduced to $NO_2^-$
by cadmium reduction and then extracted using the azo dye method described above. 2) Dissolved
$NO_2^-$ is converted to Sudan-1 and Sudan-1 is collected via solid-phase extraction. The sample is
then purified by HPLC and derivatized before analysis by GC/MS (Clark et al., 2007). Similarly,
$NO_3^-$ can be reduced to $NO_2^-$ by cadmium prior to conversion to Sudan-1 for nitrogen isotope
analysis. 3) $NO_2^-$ can be converted to $N_2$ with sulfamic acid and subsequently measured by isotope
ratio mass spectrometry (Dalsgaard et al., 2012; Bristow et al., 2016). 4) $NO_2^-$ can also be
converted into $N_2O$ by the azide method and subsequently measured by isotope ratio mass
spectrometry (Mcilvin and Altabet, 2005). The N isotopes of $NO_2^-$ and $NO_3^-$ can be measured via
the denitrifier method (Sigman et al., 2001; Weigand et al., 2016) where both $NO_2^-$ and $NO_3^-$ are
converted into $N_2O$. In addition, the $\delta^{15}N$ of $NO_3^-$ alone can be measured using the denitrifier
method after removing $NO_2^-$ with sulfamic acid (Granger and Sigman, 2009). The azide and
denitrifier methods require smaller sample volumes and offer a higher sensitivity of nitrogen
isotope detection.
Many factors may complicate the interpretation of rate measurements, e.g., isotope dilution by
regeneration of the $^{15}N$-labeled substrates and stimulation of nitrification by substrate addition
(Lipschultz, 2008). For instance, the amount of tracer addition varied substantially from <10 nM
to 5 μM, enriching the ambient pool by <10% to over 1000%. The excess addition of substrates
will likely enhance the nitrification rate, which will then reflect a potential rate instead of an in-
situ rate. In addition, the measurement of $NO_2^-$ compared to $NO_2^- + NO_3^-$ could also lead to
variations in the estimates of the ammonia oxidation rates. Specifically, $^{15}NO_2^-$ produced from
$^{15}NH_4^+$ may be further oxidized to $^{15}NO_3^-$, especially when samples are low in $NO_2^-$ concentration.
Ammonia oxidation rate may be underestimated if only $^{15}NO_2^-$ is measured instead of measuring
both $^{15}NO_2^-$ and $^{15}NO_3^-$ (Santoro et al., 2013; Peng et al., 2015). Therefore, $NO_2^-$ carrier (to increase
the $NO_2^-$ pool and trap the produced $^{15}NO_2^-$) may be added to the sample before incubation or both
$NO_2^-$ and $NO_3^-$ should be measured after incubation when ambient $NO_2^-$ concentration is low. The
$^{15}NO_2^-$ isotope dilution method may overestimate ammonia oxidation rates because $NO_2^-$ could



also be released from phytoplankton after assimilative nitrate reduction (Lomas and Lipschultz,
2006). These confounding factors may be difficult to quantify but worth recording and reporting
in publications for the sake of comparison among studies. In addition, a variety of approaches have
been applied to calculate nitrification rates. However, some methods correct for the impact of other
processes such as the uptake of the substrates or products of nitrification on rate estimates (e.g.,
Lipschultz et al., 1986; Santoro et al., 2010) while others do not (e.g., Dugdale and Goering.,

253   1967).


Nitrification supported by organic N substrates like urea and cyanate has been observed in the Gulf
of Mexico (Kitzinger et al., 2018), Pacific (Santoro et al., 2017; Wan et al., 2021), off the east
coast of the United States (Laperriere et al., 2020; Tolar et al., 2017), and in the polar oceans
(Alonso-Saez et al., 2012; Shiozaki et al., 2021). The number of these observations remains limited
compared to ammonia oxidation. They can be included in future editions of the database (i.e., not
included in the current database) and their role in the marine N cycle deserves future investigations.

**Methods for quantifying ammonia oxidizers and nitrite oxidizers**
We summarize the primers used to quantify nitrifier abundance based on both functional genes
and 16S rRNA genes using quantitative PCR (qPCR) (Table 3). The cell abundance and biomass
can be subsequently estimated based on the gene abundance, number of genes per cell and specific
cell biomass (e.g., Kitzinger et al., 2020; Khachikyan et al., 2019). The oxidation of ammonia to
hydroxylamine is catalyzed by ammonia monooxygenase, which is partly encoded by the *amoA*
gene. Primers have been designed to quantify both bacterial and archaeal *amoA* genes (Rotthauwe
et al., 1997; Francis et al., 2005; Hornek et al., 2006; Wuchter et al., 2006; Beman et al., 2008;
Mosier and Francis, 2011; Sintes et al., 2013). Archaeal ammonia oxidizers are also separated into
different ecotypes including Water Column ecotypes A and B (WCA and WCB), which
preferentially inhabit the surface vs deep ocean, respectively, or high-ammonia concentration vs
low-ammonia concentration groups, which dominate in high ammonia vs low ammonia
concentration environments, respectively. The *nxrB* gene, which encodes the beta subunit of nitrite
oxidoreductase for nitrite oxidation, has been used to quantify *Nitrospira* (Pester et al., 2014).
However, no primers targeting *nxr* genes are available for other groups of nitrite oxidizers such as
*Nitrospina*, which is the dominant group of nitrite oxidizers in the ocean (Beman et al., 2013;



Pachiadaki et al., 2017). Primers have also been designed to quantify the 16S rRNA gene
abundance of *Thaumarchaeota*, *Nitrospira*, and *Nitrospina* (Mincer et al., 2007; Graham et al.,
2007). The abundance of nitrifiers can be useful for inferring and interpreting nitrification rates.
In addition to qPCR, amplicon sequencing and quantitative metagenomics are also useful to
determine the abundance of nitrifiers (Tolar et al., 2020; Lin et al., 2019; Satinsky et al., 2013) but
these analyses are not included in the database.

Table 3. qPCR primers commonly used to quantify nitrifier abundance in the ocean.

| Target | Name | Primer sequences (5'-3') | References |
|---|---|---|---|
| Gamma-bacterial *amoA* | amoA-1F<br>amoA-2R<br>or<br>amoA-r NEW | GGGGTTTCTACTGGTGGT<br>CCCCTCKGSAAAGCCTTCTTC<br><br>CCCCTCBGSAAAVCCTTCTTC | Rotthauwe et al., 1997<br><br>Hornek et al., 2006 |
| Water Column ecotype A (WCA) archaeal-*amoA* | Arch-amoAFA<br>Arch-amoAR | ACACCAGTTTGGYTACCWTCDGC<br>GCGGCCATCCATCTGTATGT | Beman et al., 2008;<br>Francis et al., 2005 |
| Water Column ecotype A (WCB) archaeal-*amoA* | Arch-amoAFB<br>Arch-amoAR | CATCCRATGTGGATTCCATCDTG<br>GCGGCCATCCATCTGTATGT | Beman et al., 2008;<br>Francis et al., 2005 |
| Total archaeal-*amoA* | Arch-amoAF<br>Arch-amoAR | STAATGGTCTGGCTTAGACG<br>GCGGCCATCCATCTGTATGT | Francis et al., 2005 |
| High-ammonia concentration archaeal-*amoA* | Arch-amoA-for<br>Arch-amoA-rev | CTGAYTGGGCYTGGACATC<br>TTCTTCTTTGTTGCCCAGTA | Wuchter et al., 2006 |
| Low-ammonia concentration archaeal-*amoA* | Arch-amoA-for<br>Arch-amoA-rev-New | CTGAYTGGGCYTGGACATC<br>TTCTTCTTCGTCGCCCAATA | Wuchter et al., 2006<br>Sintes et al., 2013 |
| *Thaumarchaeota* 16S rRNA | GI_751F<br>GI_956R | GTCTACCAGAACAYGTTC<br>HGGCGTTGACTCCAATTG | Mincer et al., 2007 |



| *nxr* | nxrB169F | TACATGTGGTGGAACA | Pester et al., |
| | nxrB638R | CGGTTCTGGTCRATCA | 2014 |
| *Nitrospira* 16S rRNA | Nspra-675f | GCGGTGAAATGCGTAGAKATCG | Graham et al., |
| | Nspra-746r | TCAGCGTCAGRWAYGTTCCAGAG | 2007 |
| *Nitrospina* 16S rRNA | NitSSU_130F | GGGTGAGTAACACGTGAATAA | Mincer et al., |
| | NitSSU_282R | TCAGGCCGGCTAAMCA | 2007 |


# Results and Discussion

## Summary of the database

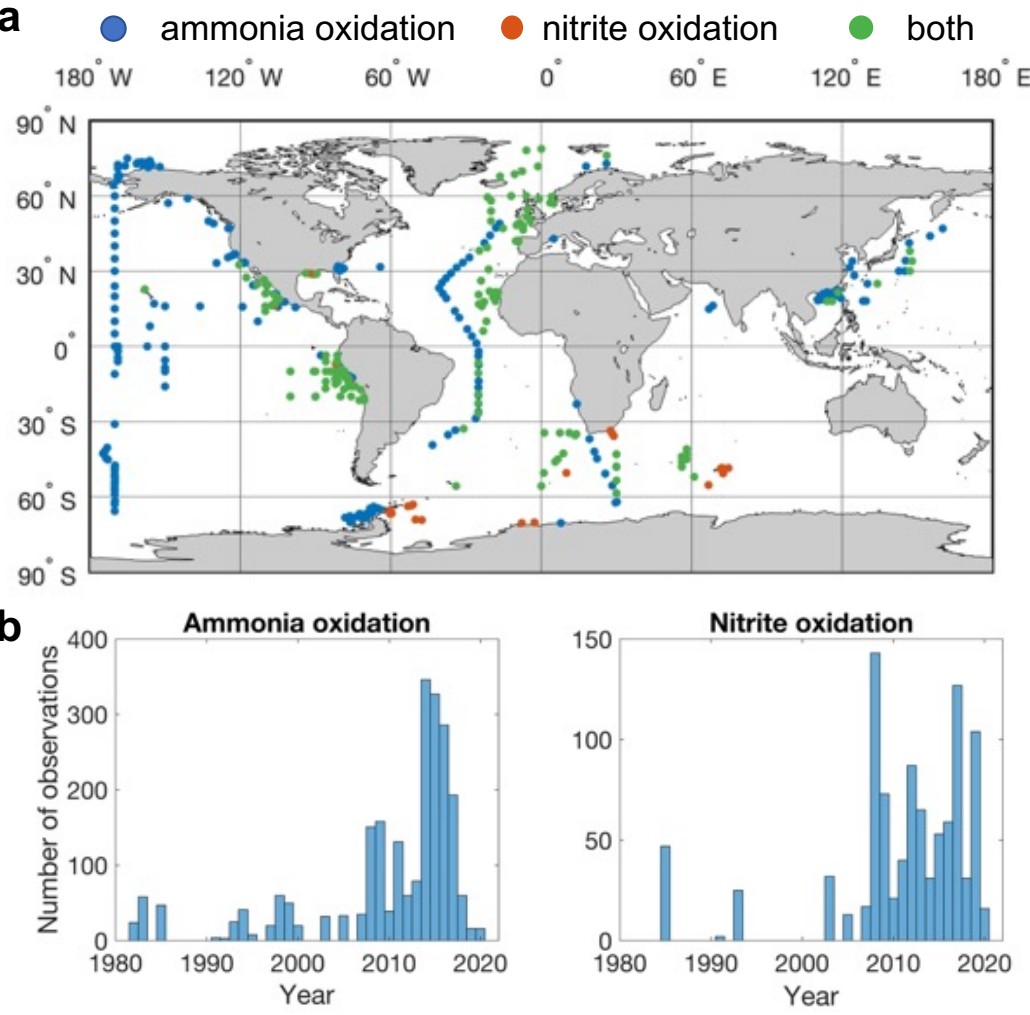


Earth System
Science Discussions
Data

Figure 2. Map showing the distribution of sampling locations for nitrification rate measurements
(a) and the number of observations each year (b). Blue points: only ammonia oxidation is
measured; red points: only nitrite oxidation is measured. Green points: both ammonia oxidation
and nitrite oxidation are measured.

In total, there are 2393 and 1006 measurements of ammonia oxidation and nitrite oxidation,
respectively (Figure 2). Ammonia oxidation and nitrite oxidation have been concurrently measured
at 418 locations. The Pacific Ocean has the largest number of nitrification observations followed
by the Atlantic Ocean, Southern Ocean and Indian Ocean. Particularly, meridional transects across
ocean basins and biomes have been conducted in the North Pacific and Atlantic (Shiozaki et al.,
2016; Clark et al., 2008; Clark et al., 2022). Observations have recently expanded into oxygen
minimum zones (Beman et al., 2012; Beman et al., 2013; Frey et al., 2020; Frey et al., 2022; Peng
et al., 2015; Peng et al., 2016; Santoro et al., 2021; Sun et al., 2017) and polar oceans (Cavagna et
al., 2015; Shiozaki et al., 2019; Smith et al., 2022; Mdutyana et al., 2022a and b; Mdutyana et al.,
2020; Flynn et al., 2021). Nitrification rates are more frequently measured after 2010 (Figure 2b).

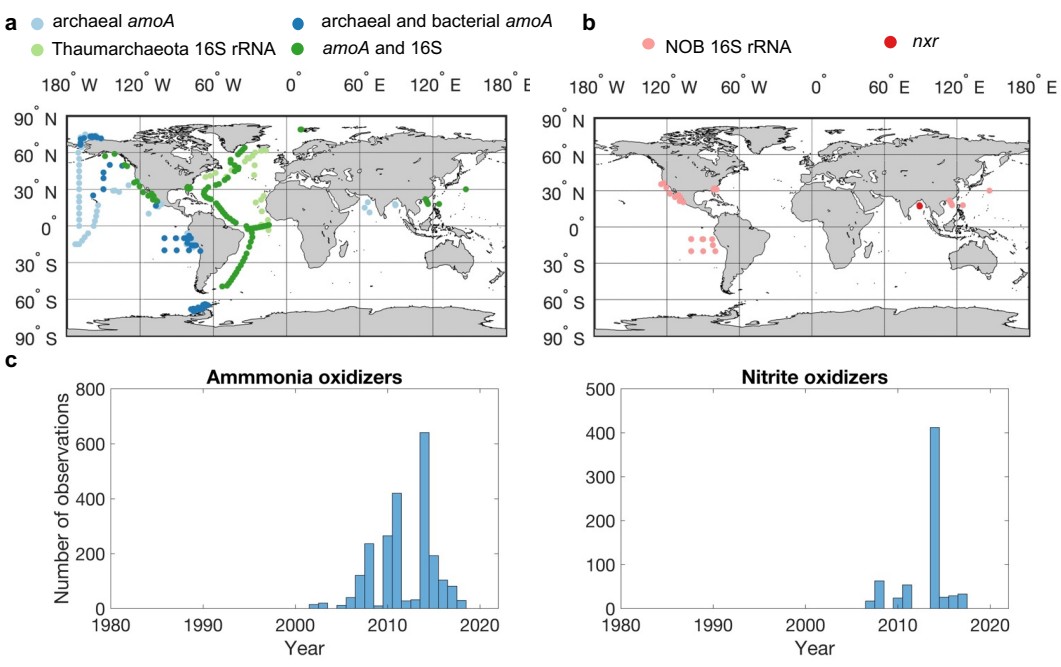




Figure 3. Maps showing the distribution of sampling locations for ammonia oxidizers (a) and
nitrite oxidizers (b), and the number of observations each year (c). (a) light blue points: only
archaeal *amoA* was quantified. Dark blue points: both archaeal and bacterial *amoA* genes were
quantified. Light green points: 16S rRNA gene of *Thaumarchaeota* was quantified; dark green
points: both archaeal *amoA* and 16S rRNA gene of *Thaumarchaeota* were quantified. (b) pink
points: 16S rRNA of nitrite oxidizers was quantified; red points: *nxr* gene of nitrite oxidizers was
quantified.

In total, there are 2187 and 631 measurements of ammonia oxidizer and nitrite oxidizer abundance,
respectively (Figure 3). Most of the nitrifier quantifications have been conducted in the tropical
and subtropical oceans (Figure 4a). Data are sparse in the central Pacific, Indian Ocean and
Southern Ocean (with the exception of the West Antarctic Peninsula). Both archaeal *amoA* and
16S rRNA genes of *Thaumarchaeota* were quantified on a transect across the Atlantic (Sintes et
al., 2016). There are far fewer observations of nitrite oxidizers compared to ammonia oxidizers.
Notably, there are only 27 observations of *nxr* genes. The number of the quantification of nitrifier
abundance starts to accumulate since 2002 (Figure 3c). Most of the observations of nitrite oxidizers
originates from one study where samples were collected in 2014 (Liu et al., 2018). Nitrification
rate and nitrifier abundance are sometimes determined at the same location, which allows us to
assess the relationship between biogeochemical rate and the abundance of functional groups (e.g.,
Peng et al., 2015; Shiozaki et al., 2019; Santoro et al., 2021).

**Distribution of ammonia oxidation**
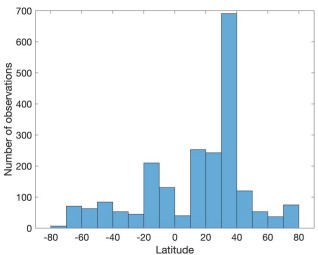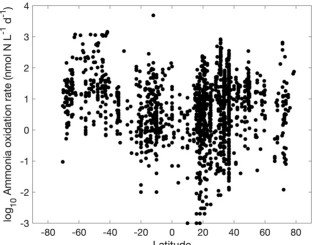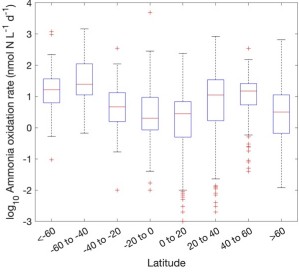
Figure 4. Number of ammonia oxidation observations and ammonia oxidation rates within
latitudinal bands. For the boxplot in this figure and figures throughout the manuscript, the red line
in each box is the median. The bottom and top of each box are the 25th and 75th percentiles of the





observations, respectively. The error bars represent 1.5 times the interquartile range away from the
bottom or top of the box, with red + signs showing outliers beyond that range.

A large number of observations exist for the tropical and temperate oceans (Figure 4), particularly
in the 30-40°N band where rates were measured in offshore waters of Georgia and California
(Tolar et al., 2020; Liu et al., 2018). Ammonia oxidation rates vary from <0.01 to over 1000 nmol
N L$^{-1}$ d$^{-1}$ with a median value of 7.7±9.8 nmol N L$^{-1}$ d$^{-1}$. There is no clear latitudinal trend in the
ammonia oxidation rates. In contrast, Clark et al. (2022) found higher ammonia oxidation rates in
the southern hemisphere along the north-south transect in the Atlantic Ocean. This latitudinal
pattern is hypothesized to be explained by the difference in the supply of dissolved organic
nitrogen (DON) by lateral transport into the gyre interior from the eastern boundary upwelling
(Clark et al., 2022). The stimulation of ammonia oxidation rates by a lateral DON supply has also
been observed in the Western Pacific (Xu et al., 2018).

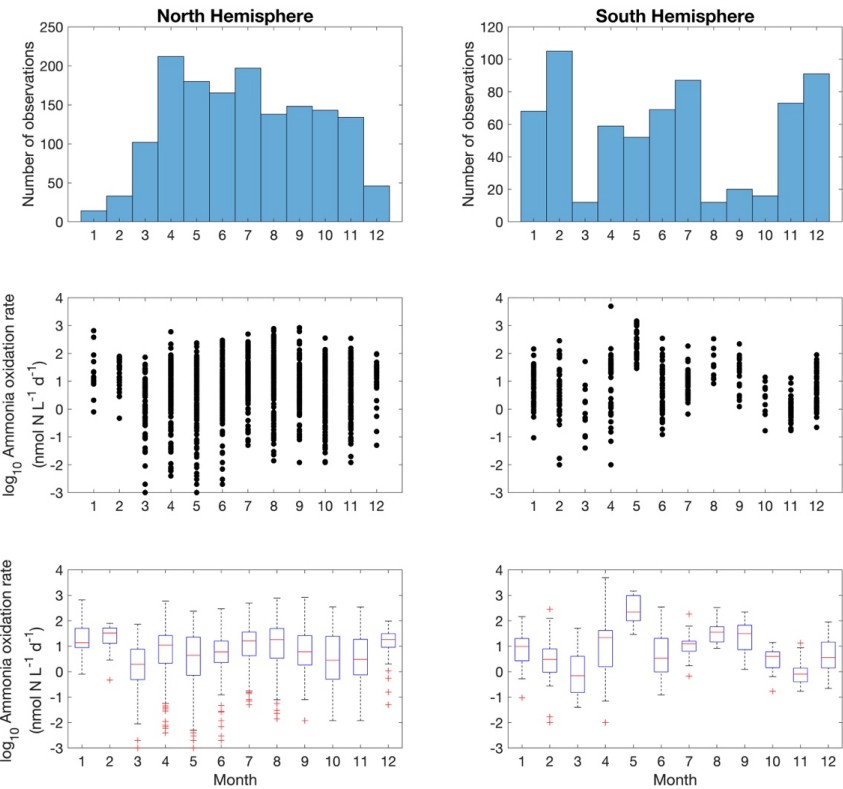






Figure 5. Monthly variation of ammonia oxidation observations and ammonia oxidation rates
divided into observations taken in the Northern Hemisphere (left panels) and Southern Hemisphere
(right panels).

More ammonia oxidation measurements were conducted in summer in both hemispheres (Figure
5) which is likely due to the more challenging weather conditions in winter for field explorations.
The northern hemisphere has more observations compared to the southern hemisphere. Although
no clear seasonal pattern is apparent for ammonia oxidation rates at a global scale, seasonal
variation in ammonia oxidation has been seen at time-series stations near and offshore of
California (Ward, 2005; Tolar et al., 2020; Laperriere et al., 2020). In addition, ammonia oxidation
showed a substantial seasonal pattern in the polar ocean with higher rates observed in the $NH_4^+$-
enriched dark winter season (Baer et al., 2017; Mdutyana et al., 2020; Mdutyana et al., 2022b).

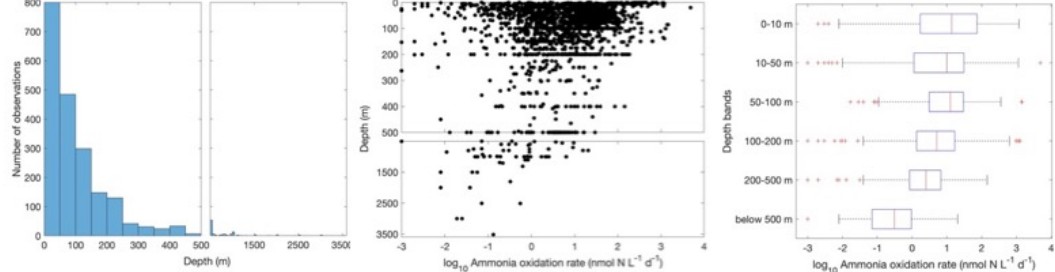


Figure 6. Vertical variation of ammonia oxidation observations and ammonia oxidation rates.

Most of the ammonia oxidation rate measurements were made shallower than 500 m, accounting
for ~96% of the total measurements (Figure 6). Ammonia oxidation rates often reach a maximum
near the base of the euphotic zone or in the 50-100 m layer before decreasing with depth below
the euphotic zone. Although nitrification is thought to be inhibited by light, high ammonia
oxidation rates >100 nmol N $L^{-1}$ $d^{-1}$ have been observed within the euphotic zone (Raes et al.,
2020; Bianchi et al., 1997), suggesting complex regulation of nitrification in the surface ocean.
This complicates the interpretation of the source of $NO_3^-$ in the euphotic zone and further the $NO_3^-$
-supported new production (Diaz and Raimbault, 2000; Yool et al., 2007; Grundle et al., 2013;
Mdutyana et al. 2020).




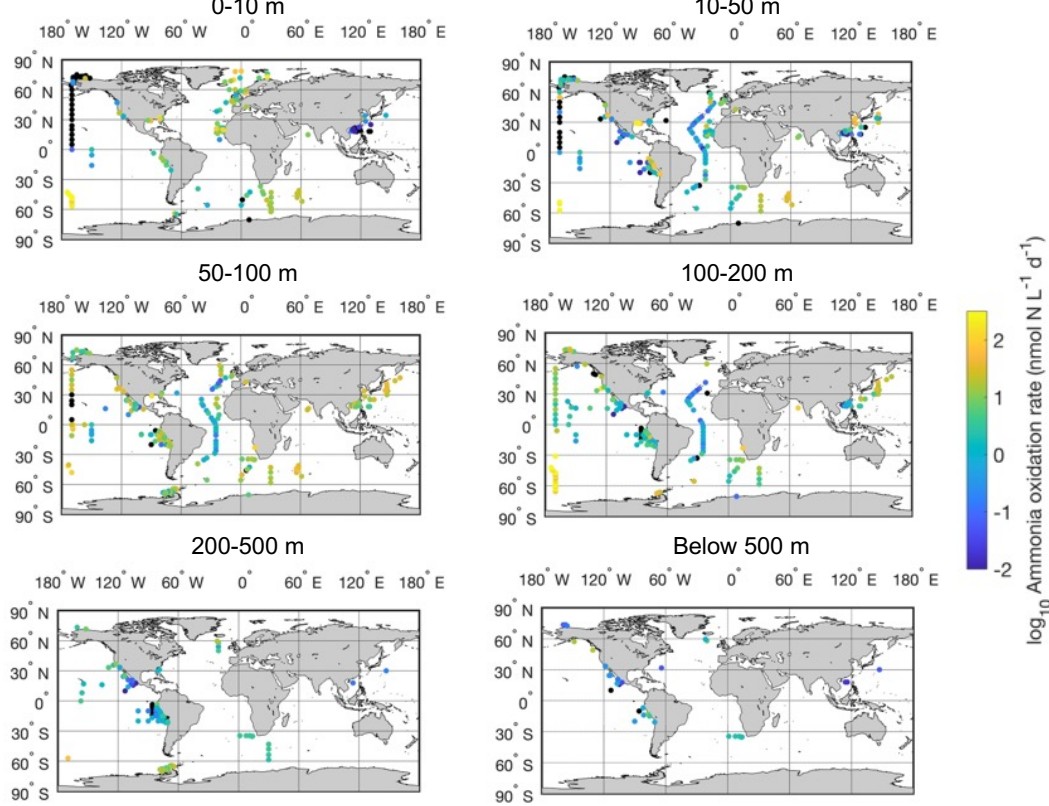


Figure 7. Spatial distribution of ammonia oxidation rates in six depth intervals in the global ocean.

Locations with rates below the detection limit are shown in the black circles.


There is a large spatial and vertical variation in ammonia oxidation rates (Figure 7). Some hotspots

with rates >100 nmol N L$^{-1}$ d$^{-1}$ include the subpolar North Atlantic (Clark et al., unpublished),

Southern Ocean (Mdutyana et al., 2020), and coastal waters off California and Georgia (Tolar et

al., 2020; Liu et al., 2018). Particularly, there are extremely high ammonia oxidation rates >1000

nmol N L$^{-1}$ d$^{-1}$ observed in the surface Pacific Southern Ocean (Raes et al., 2020), deserving further

studies to confirm this pattern. In contrast, some low rates <0.01 nmol N L$^{-1}$ d$^{-1}$ or rates below the

detection limit are found in the surface sunlit North Pacific, which is likely caused by the light

inhibition on nitrifiers, and nitrifiers' competition with phytoplankton for $NH_4^+$ in well-lit areas

(Smith et al., 2014). For example, peak ammonia oxidation rates are often found in regions/depths

where $NO_3^-$ is present or light levels are low such that competition of nitrifiers with phytoplankton
for $NH_4^+$ diminishes (Figure 8; Wan et al., 2021). Additionally, low rates are found in oxygen-
depleted waters of the eastern tropical Pacific where ammonia oxidation is likely limited by
oxygen availability (Peng et al., 2016)

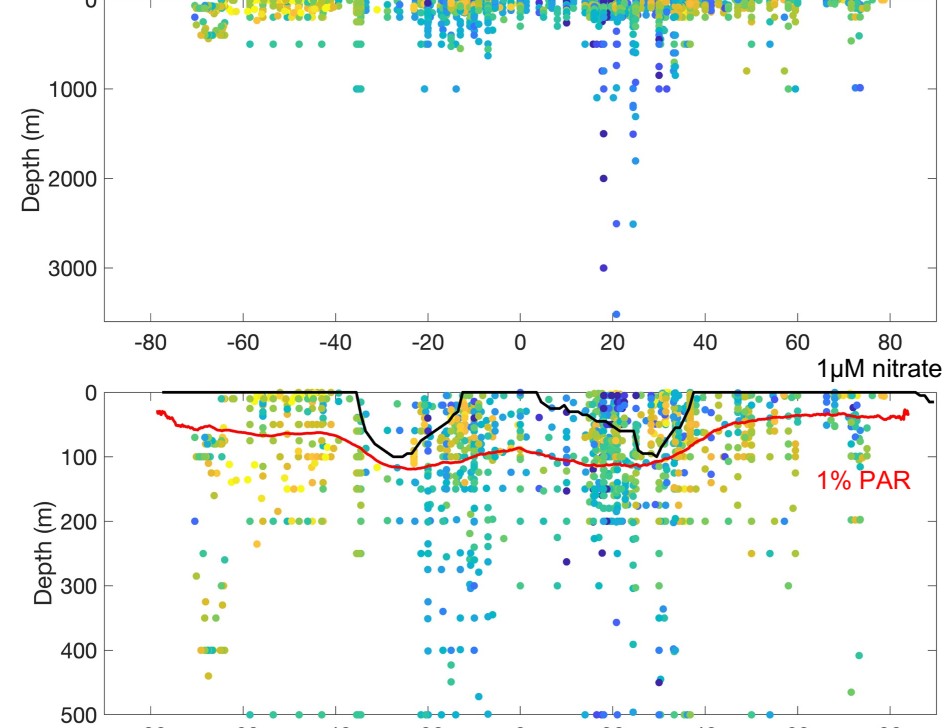


Figure 8. Latitudinal and vertical distribution of ammonia oxidation rates. The lower panel shows
data from the top 500 m. The climatological depths of the euphotic zone (1% PAR) obtained from
MODIS satellite observations and 1 μM nitrate obtained from World Ocean Atlas 2018 (García et
al., 2019) are shown by the red line and black lines, respectively.

**Distribution of nitrite oxidation**

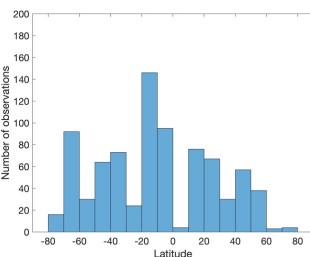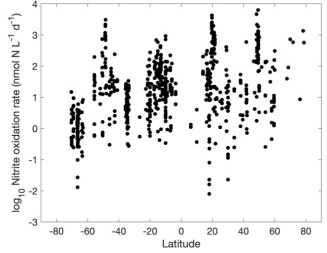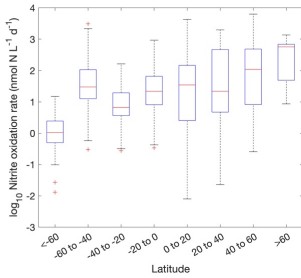

Figure 9. Number of nitrite oxidation observations and nitrite oxidation rates within latitudinal bands.

Similar to ammonia oxidation, the majority of the nitrite oxidation observations were conducted in the tropical and subtropical oceans (Figure 9), particularly in the eastern tropical Pacific oxygen minimum zones (Ward et al., 1989; Peng et al., 2015; Kalvelage et al., 2013; Santoro et al., 2021). Recent observations extended into the Southern Ocean (Cavagna et al., 2015; Mdutyana et al., 2020; Mdutyana et al., 2022a; Flynn et al., 2021). The rates vary from 0.01 to >1000 nmol N L$^{-1}$ d$^{-1}$ with a median value at 15.9±10.7 nmol N L$^{-1}$ d$^{-1}$. Nitrite oxidation rates seem to increase from the southern hemisphere to northern hemisphere. The lowest median rates were found in the Southern Ocean south of 60°S, which is hypothesized to be regulated by low iron availability (Mdutyana et al., 2022a). Overall, more measurements of nitrite oxidation over a large spatial scale are desired to resolve the latitudinal distribution of nitrite oxidation rates.

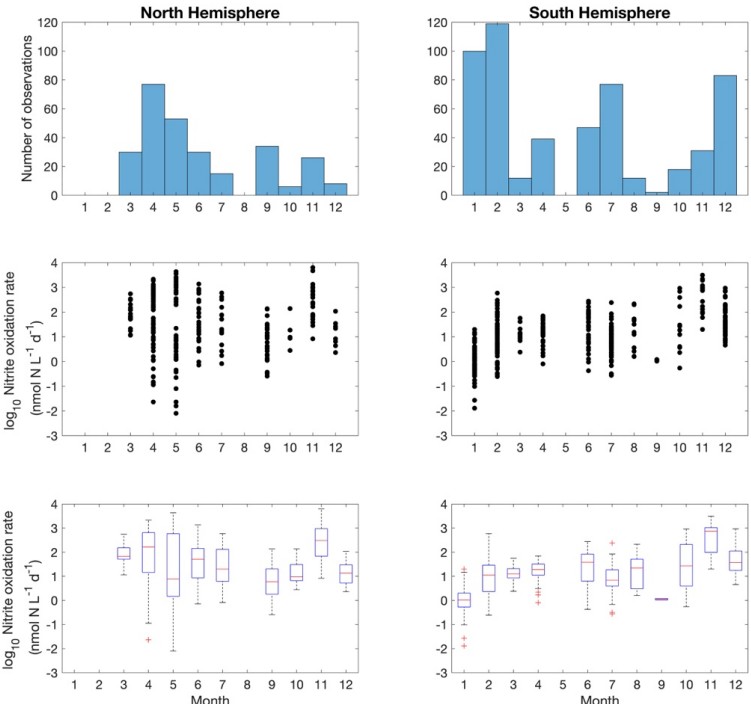


Figure 10. Monthly variation of nitrite oxidation observations and nitrite oxidation rates.

Nitrite oxidation measurements are limited in winter in the northern hemisphere (Figure 10). No
clear seasonal pattern is found for nitrite oxidation rates at a global scale, except for some of the
lowest rates detected in January in the Southern Ocean (austral summer). In addition to iron
limitation, light inhibition and competition with phytoplankton for nitrite during the growing
season may be important factors driving these low rates. Unlike ammonia oxidation, there is no
time-series study of nitrite oxidation to show its seasonal variations.

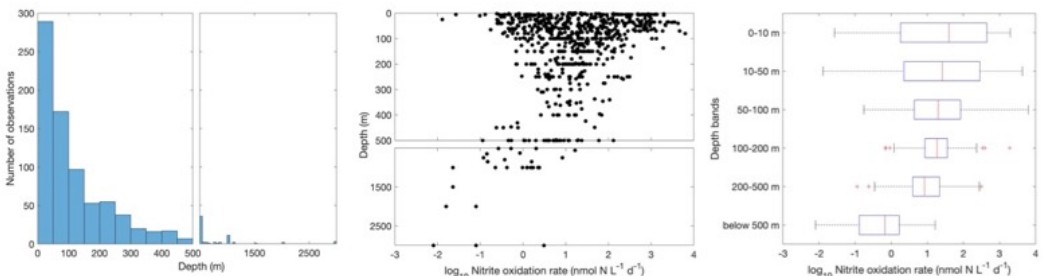




Figure 11. Vertical variation of nitrite oxidation observations and nitrite oxidation rates.

Most of the nitrite oxidation rates were also measured at depths shallower than 500 m, accounting for ~94% of the total measurements (Figure 11). There is a large spatial variation in the nitrite oxidation observations and rates (Figure 12). Observations are lacking in the central Pacific Ocean and Indian Ocean outside of the oxygen minimum zones. Nitrite oxidation rates decrease with depth. Globally, the highest median nitrite oxidation rates were found in the surface water (0-10 m layer), which is mainly attributed to the high surface rates observed over the United Kingdom shelves, subpolar North Atlantic and Mauritanian upwelling system (Figure 12; Clark et al., unpublished; Clark et al., 2016). After removing these high surface nitrite oxidation rates, the depth profiles of nitrite oxidation often show a subsurface maximum that is slightly deeper than the subsurface maximum of ammonia oxidation (Figure 13). This difference may be related to the higher sensitivity of nitrite oxidizers/nitrite oxidation to light (Wan et al., 2021; Olson, 1981b). Interestingly, some deep peaks of nitrite oxidation rates have been found in the oxygen-depleted waters in the oxygen minimum zones (Peng et al., 2015; Babbin et al., 2020; Ward et al., 1989; Beman et al., 2013). These high rates stand out in depths below the 1 μM nitrate threshold and above the 1% PAR level between 20°N and 20°S (Figure 14). Many hypotheses (Sun et al., 2023) have been proposed to explain the observed "anaerobic" nitrite oxidation, including alternative oxidants like iodate (Babbin et al., 2017), distinct nitrite oxidizers that are only present in the OMZs and adapted to the low oxygen conditions (Sun et al., 2021), nitrite dismutation ($2H^+ + 5NO_2^- \rightarrow N_2 + 3NO_3^- + H_2O$; van de Leemput et al., 2011; Babbin et al., 2020; Tracey et al., 2022), and oxygen intrusions (Buchanan et al., 2023). Whether nitrite oxidation is truly anaerobic and how nitrite oxidation is sustained in oxygen depleted waters remain to be determined.

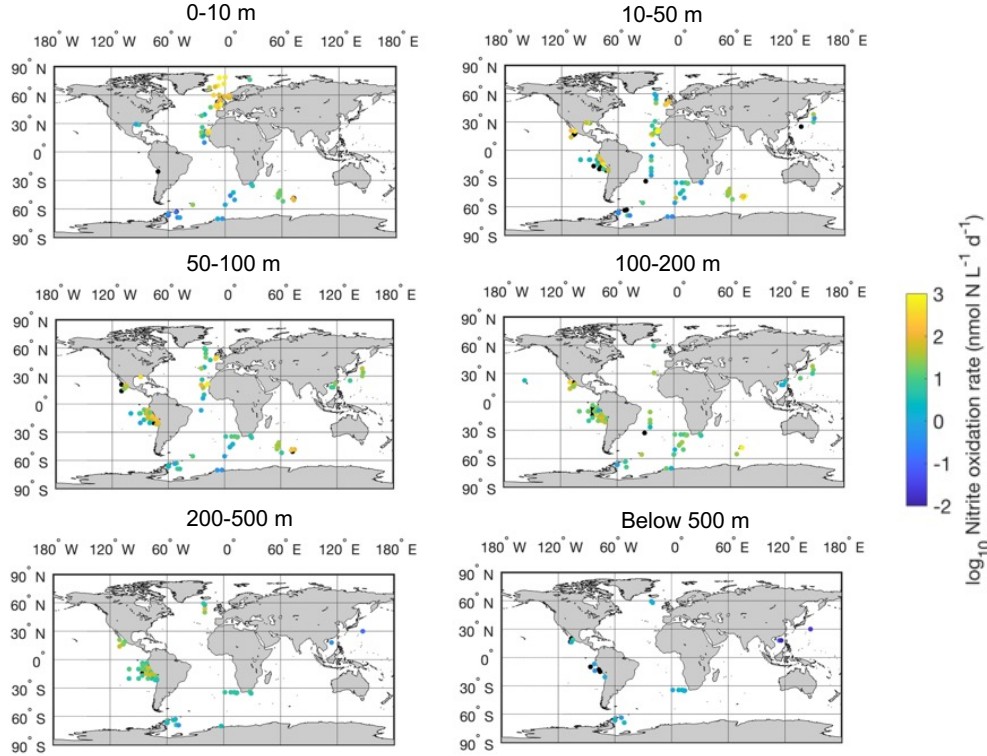


Figure 12. Spatial distribution of nitrite oxidation rates in six depth intervals in the global ocean.
Locations with rates below the detection limit are shown in the black circles.

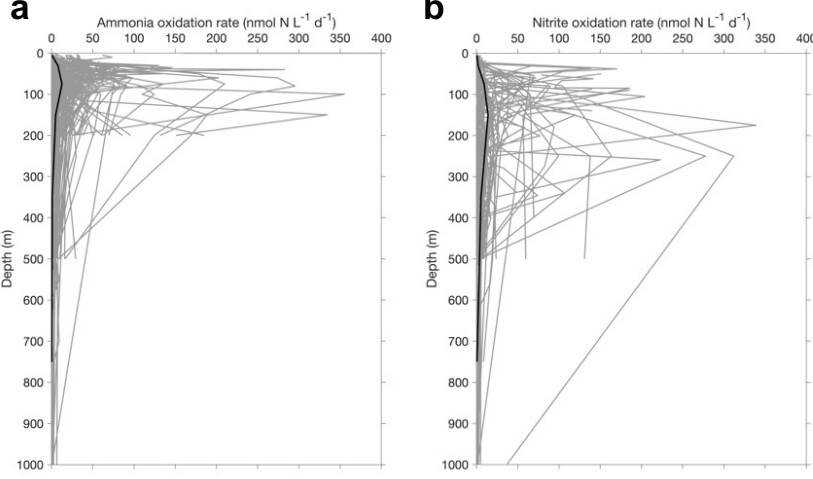


Figure 13. Depth profiles of ammonia oxidation (a) and nitrite oxidation (b) in the top 1000 m. Only depth profiles with five or more measurements are included in this figure. The median profiles of ammonia oxidation and nitrite oxidation are shown in thick black lines, showing the maximum of nitrite oxidation deeper than the maximum of ammonia oxidation.

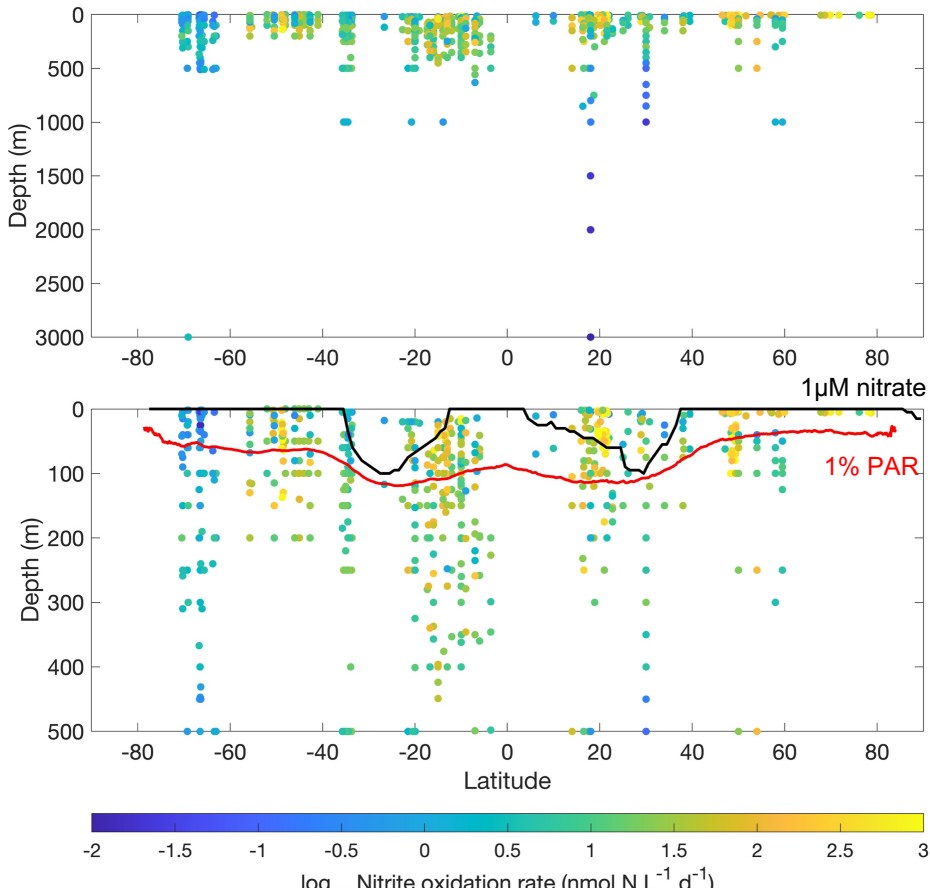

Figure 14. Latitudinal and vertical distribution of nitrite oxidation rates. The lower panel shows data from the top 500 m. The climatological depth of the euphotic zone (1% PAR) and 1 μM nitrate are shown by the red and black lines respectively.

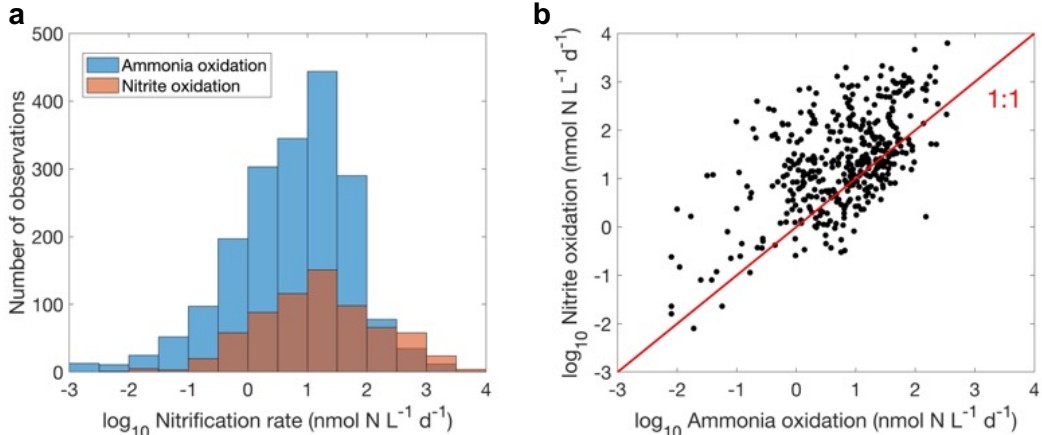


Figure 15. Comparison between ammonia oxidation and nitrite oxidation. (a) Histogram of both
rates globally. (b) Relationship between ammonia oxidation and nitrite oxidation measured at the
same locations and time (y=0.62*x+0.82, r=0.54, p<0.01).

Overall, there are fewer nitrite oxidation rate measurements compared to ammonia oxidation
measurements (Figure 15a). Ammonia oxidation and nitrite oxidation are generally of similar
magnitude (Figure 15b), leading to the low concentration of $NO_2^-$ in most of the ocean. However,
ammonia oxidation and nitrite oxidation could be decoupled. For example, higher ammonia
oxidation rates than nitrite oxidation rates (Lomas and Lipschultz, 2006) and competition between
ammonia oxidation and phytoplankton ammonium assimilation (Zakem et al. 2018) may both
partly explain the presence of the primary nitrite maximum The median nitrite oxidation rate is
higher than the median ammonia oxidation rate (15.9 vs 7.7 nmol N L$^{-1}$ d$^{-1}$), which may be related
to nitrite production pathways from urea and cyanate oxidation in addition to ammonia oxidation
(Wan et al., 2022; Kitzinger et al., 2018). Consistently, when comparing ammonia oxidation and
nitrite oxidation rates measured at the same locations and same time, nitrite oxidation rates are
mostly higher (Figure 15b). Mechanisms driving the decoupling of ammonia oxidation and nitrite
oxidation deserve further investigations.



## Distribution of ammonia oxidizers

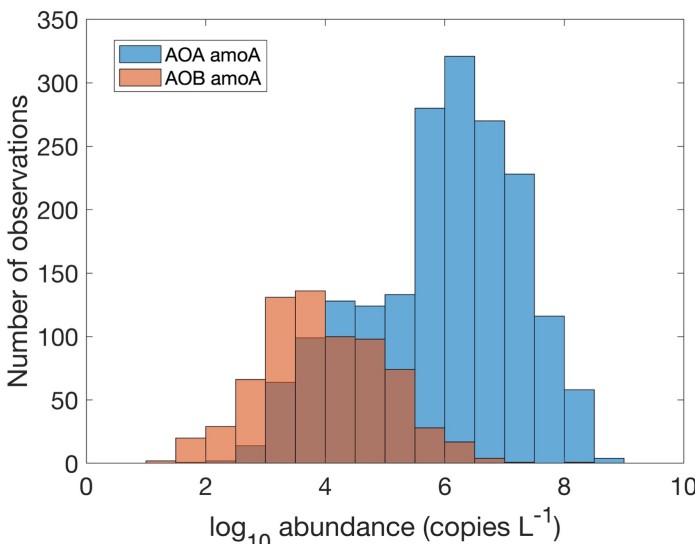

Figure 16. Comparison between the gene abundance of AOA *amoA* and AOB *amoA*.

Figure 17. Comparison between archaeal *amoA* and *Thaumarchaeota* 16S rRNA gene abundances
(y=0.78*x+1.66, r=0.9, p<0.01).



There are 1893, 892, 1073 measurements of the abundance of archaeal *amoA* gene, bacterial *amoA*
and 16S rRNA of *Thaumarchaeota*, respectively. In addition, 1204 and 1101 measurements were
separately conducted for water column ecotype A (WCA) *amoA* and water column ecotype B
(WCB) *amoA*. The AOA *amoA* abundance with median of 1.34 x $10^6$ copies $L^{-1}$ is substantially
higher than AOB *amoA* gene abundance with median of 7.96 x $10^3$ copies $L^{-1}$ (Figure 16),
confirming the dominance of archaeal ammonia oxidizers in the ocean. We also found that
*Thaumarchaeota* 16S rRNA gene abundance positively correlates with but slightly outnumbers
the *amoA* gene abundance (Figure 17). This may suggest that not all the *Thaumarchaeota* contain
the *amoA* genes to oxidize $NH_4^+$ or some organisms containing *amoA* genes (such as the
*Nitrosopumilus*-like group) may have been missed due to primer bias (Sintes et al., 2016; Hiraoka
et al., preprint), Since archaeal *amoA* genes have the largest number of observations and better
represent ammonia oxidation capability, we will use it to show the spatial and vertical distribution
of ammonia oxidizer abundance.

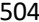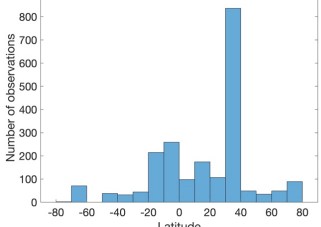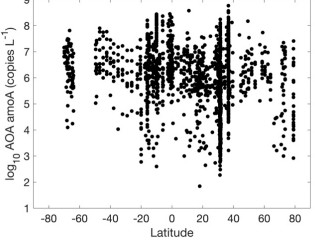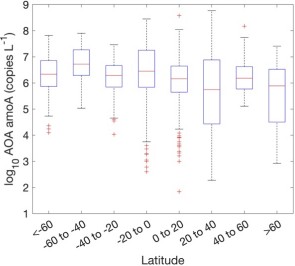


Figure 18. Number of observations of ammonia oxidizers and the abundance of ammonia oxidizers
within latitudinal bands.

The eastern Pacific Ocean and Atlantic Ocean have the majority of the observations for ammonia
oxidizers, particularly in the 30-40°N band where ammonia oxidizers were measured in the coastal
waters off California and Georgia (Liu et al., 2018; Tolar et al., 2020). In contrast, observations in
the Indian Ocean and Southern Ocean are scarce. The AOA *amoA* gene abundance varies from a
few copies per liter in the surface ocean to over $10^8$ copies $L^{-1}$ in the subsurface of equatorial
Atlantic. There is no clear latitudinal trend in the abundance of ammonia oxidizers.



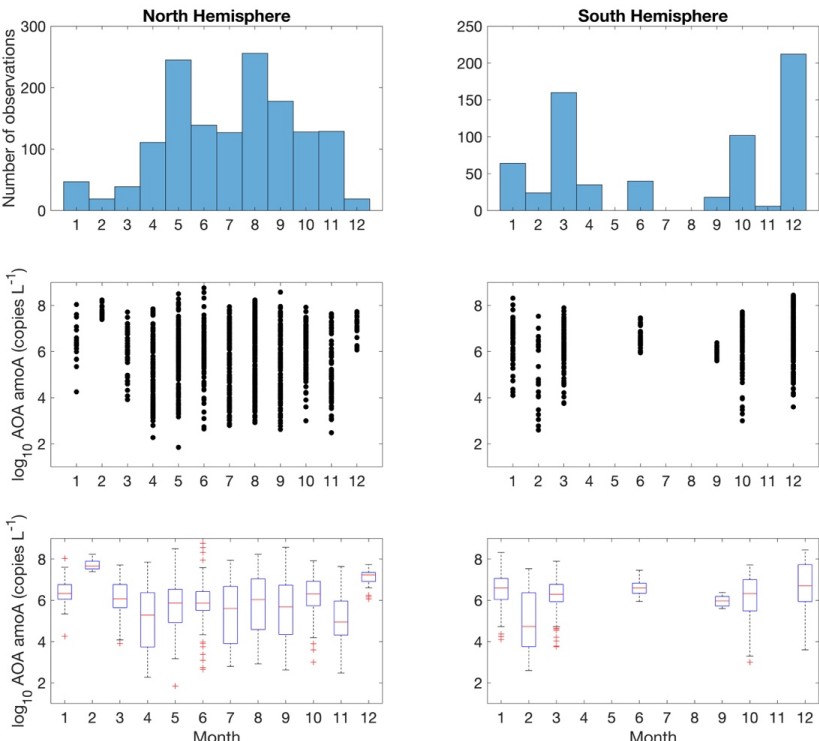

Figure 19. Monthly variation of the observation and abundance of ammonia oxidizers.
There are substantially more observations in the northern hemisphere than the southern
hemisphere. We do not see a clear seasonal trend in nitrifier abundance due to the large monthly
variation. A time-series study in the Monterey Bay shows that seasonality can be observed for the
top 200 m while the overall community of ammonia oxidizers was stable at 500 m (Tolar et al.,
2020). In addition, mid-summer peaks in *Thaumarchaeota* abundance have been observed at the
coast off Georgia (Hollibaugh et al., 2013). More time-series studies with high-frequency sampling
would be useful for characterizing the response of the nitrifier community to seasonal changes in
environmental drivers.

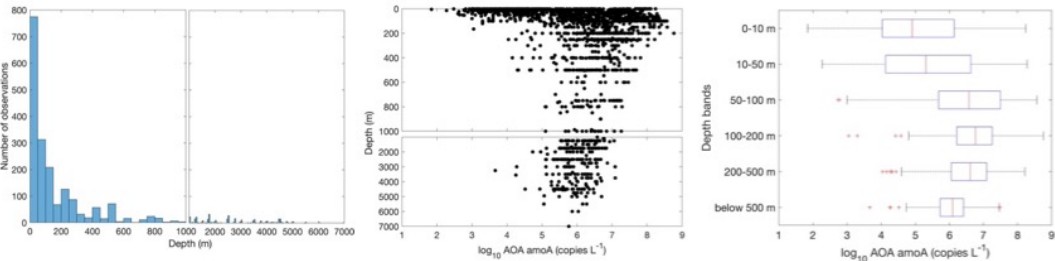


Figure 20. Vertical distribution of archaeal *amoA* observations and archaeal *amoA* gene abundance.


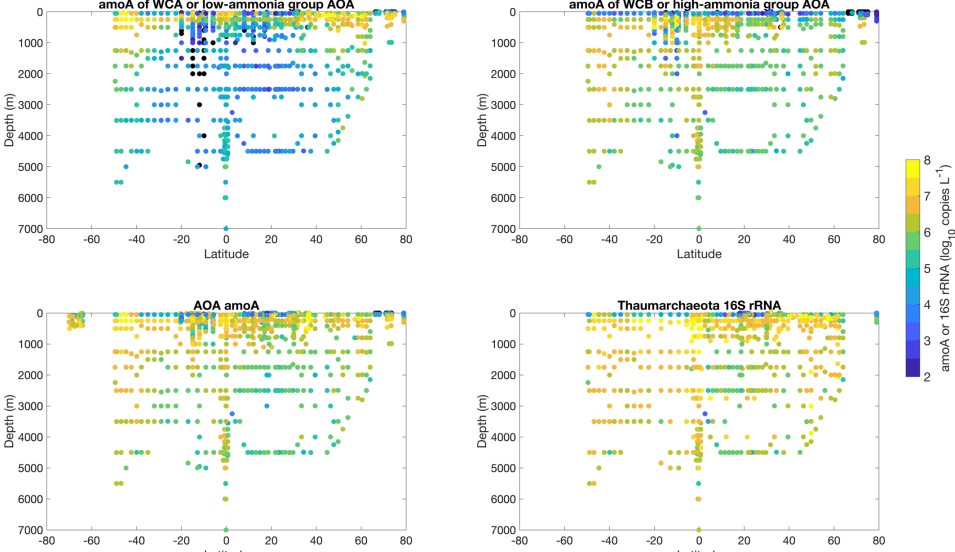


Figure 21. Vertical distribution of archaeal *amoA* gene abundance and 16S rRNA of *Thaumarchaeota* across the latitudinal gradient. WCA and high-ammonia concentration groups are shown together while WCB and low-ammonia concentration groups are shown together.


Most of the abundance measurements of ammonia oxidizers were made in the top 1000 m (Figure 20). Median ammonia oxidizer abundance increases from $\sim 10^5$ copies $L^{-1}$ in the 0-10 m depth layer to $\sim 10^7$ copies $L^{-1}$ in the 100-200 m layer, then decreases with depth and remains relatively constant at $\sim 10^6$ copies $L^{-1}$ in the deep ocean below 500 m depth. The archaeal *amoA* is sometimes quantified separately for two ecotypes including water column groups A and B. Water column

group A dominates the upper 200 meter while water column group B is more abundant in the
mesopelagic and bathypelagic deep ocean below 500 m, likely reflecting their different affinities
for $NH_4^+$ (Beman et al., 2008; Sintes et al., 2016). The vertical distribution of ammonia oxidizers
is similar to the vertical distribution of ammonia oxidation rates (Figure 13).

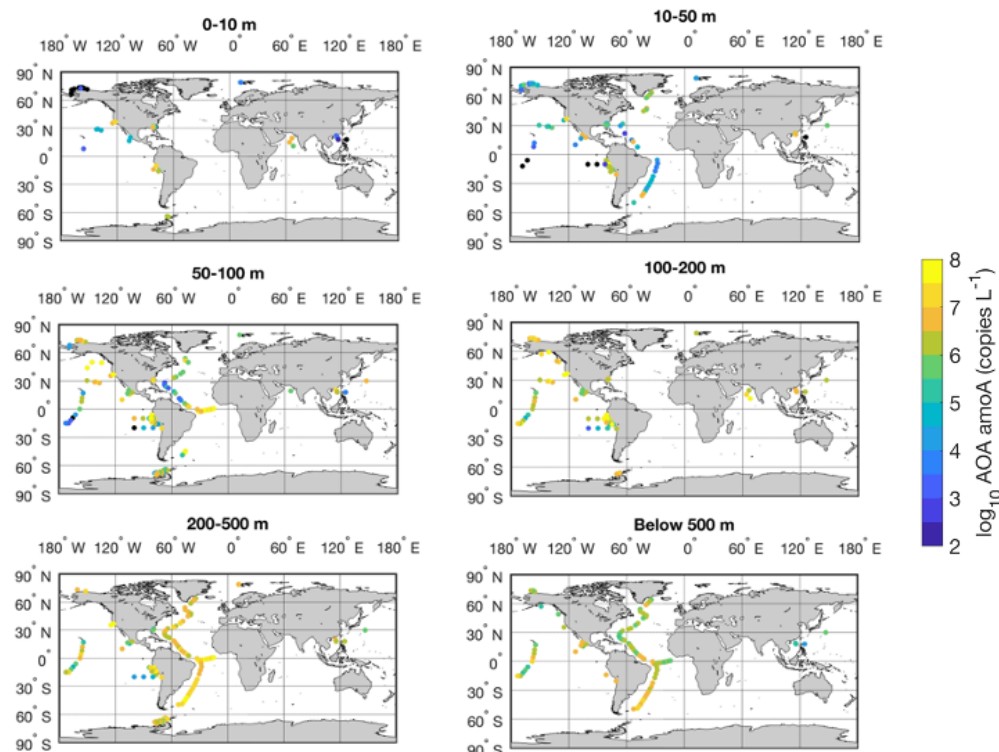


Figure 22. Spatial distribution of *amoA* gene in six depth intervals in the global ocean. Locations
with abundance below the detection limit are shown in the black circles.

There is a large spatial variation in the abundance of ammonia oxidizers (Figure 22). High
abundances are found in the tropical Atlantic and eastern tropical Pacific where upwelling drives
high rates of marine primary production. In contrast, some of the lowest abundances of ammonia
oxidizers are found in the South China Sea and oligotrophic subtropical Pacific. Therefore, the
distribution of marine productivity and organic matter production and export may play an
important role in regulating the distribution of ammonia oxidizers because ammonia oxidizers rely
on the supply of $NH_4^+$, which is generated by of organic matter decomposition.




**Distribution of nitrite oxidizer abundance**

There are only seven studies available reporting the abundance of nitrite oxidizers in the ocean. One study used the *nxr* marker gene and the other six studies used 16S rRNA gene of either *Nitrospina* or *Nitrospira*. Since *Nitrospina* is the dominant nitrite oxidizer in the ocean (Beman et al., 2013; Pachiadaki et al., 2017) and accounts for most of the observations, we use it to show the distribution of nitrite oxidizers.


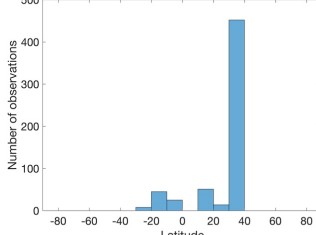 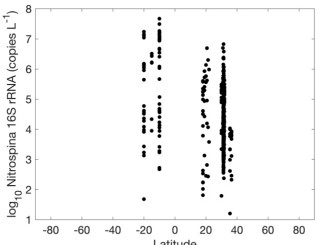 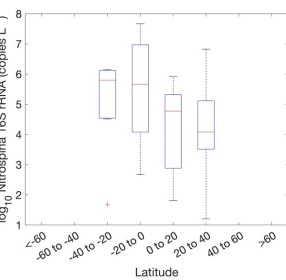


Figure 23. Number of observations and abundance of *Nitrospina* within latitudinal bands.


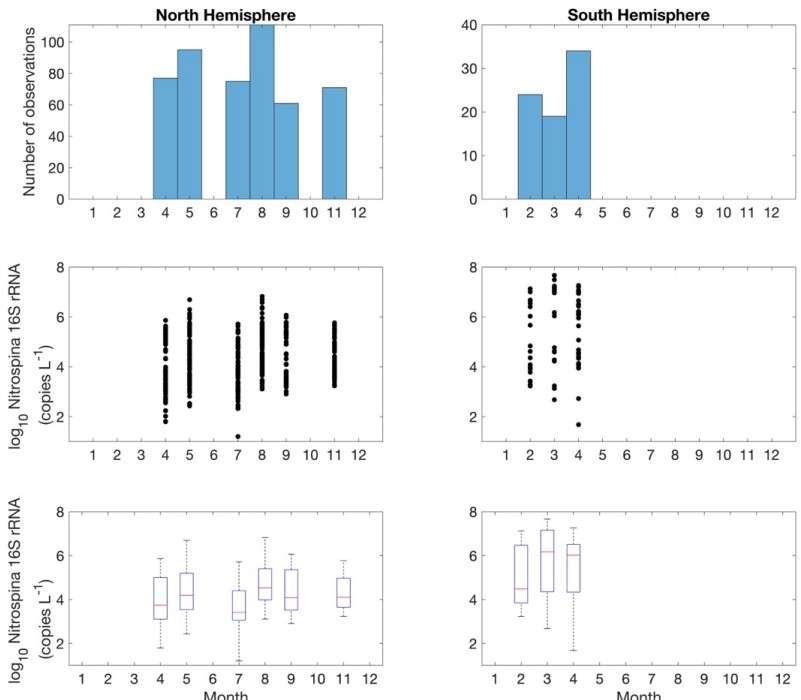


Figure 24. Monthly variation of observations and abundance of *Nitrospina*.

Quantification of nitrite oxidizers using the 16S rRNA gene is limited to a few locations between
40ºN to 40ºS including the coastal waters off California and Georgia, the eastern tropical South
Pacific, Bay of Bengal, and western Pacific (Figure 23). The number of observations is dominated
by one study conducted near the coast of Georgia (Liu et al., 2018). The highest abundance of 4.68
x $10^7$ copies $L^{-1}$ was found in the eastern tropical South Pacific. No clear latitudinal or seasonal
trend can be determined based on the limited number of observations (Figures 23-24).

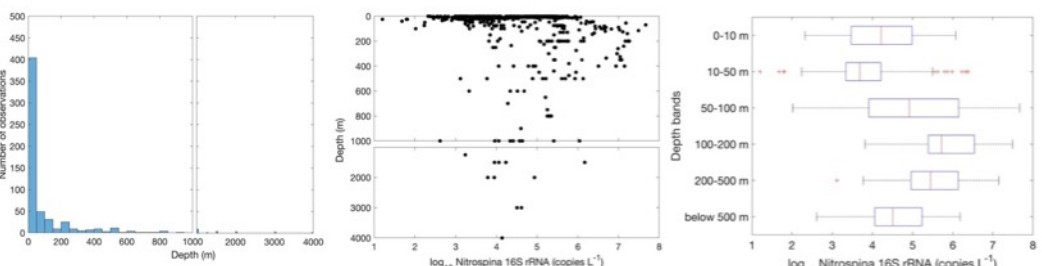

Figure 25. Vertical distribution of *Nitrospina* observations and gene abundance.

The vertical distribution of nitrite oxidizers resembles the vertical distribution of ammonia
oxidizers: increases from ~$10^4$ copies L$^{-1}$ in the surface 0-10 m depth layer to a maximum of ~$10^6$
copies L$^{-1}$ in the 100-200 m layer, then decreases to ~$10^{4.5}$ copies L$^{-1}$ in the deep ocean below 500
m (Figures 25-26). However, data below 500 m are insufficient to describe the distribution of
nitrite oxidizers in the deep ocean. The vertical distribution of nitrite oxidizers qualitatively
matches the vertical distribution of nitrite oxidation rates (Figure 13).

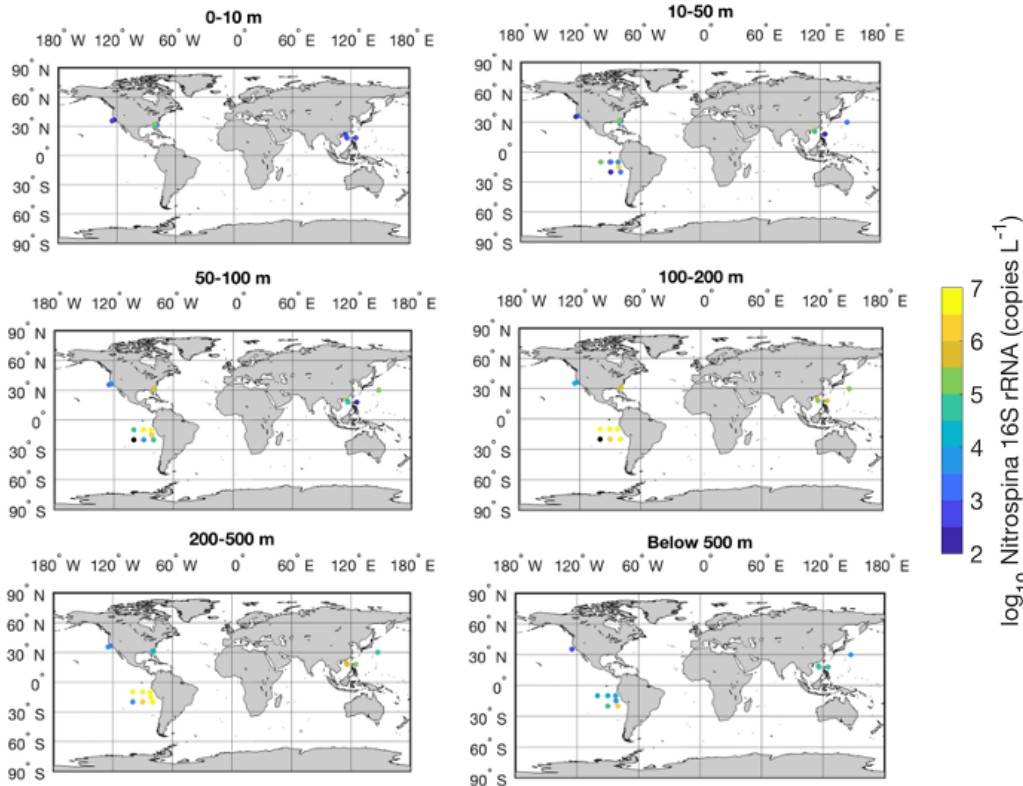


Figure 26. Spatial distribution of *Nitrospina* in six depth intervals in the global ocean. Locations
with abundances below the detection limit are shown in the black circles.


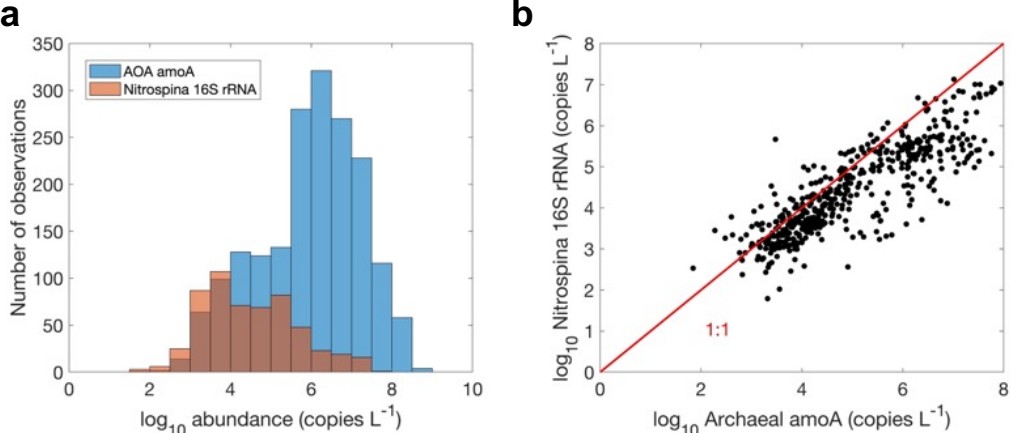


Figure 27. Comparison between the abundance of the archaeal *amoA* gene and *Nitrospina* 16S
rRNA gene (y=0.68*x+1, r=0.85, p<0.01).

When comparing ammonia oxidizers with nitrite oxidizers, median abundance of ammonia
oxidizers of $1.34 \times 10^6$ copies $L^{-1}$ is approximately two orders of magnitude higher than the median
nitrite oxidizer of $2.14 \times 10^4$ copies $L^{-1}$. The difference in their abundance has been predicted by
the relative biomass yields and cell quotas (Zakem et al., 2018; Zakem et al., 2022) and
alternatively is explained by the difference in the mortality/loss rates between AOA and *Nitrospina*
(Kitzinger et al., 2020). In addition, there is a positive relationship between the abundance of
ammonia oxidizers and nitrite oxidizers (Figure 27) as previously shown in observations from the
Pacific (Santoro et al., 2019), indicating their coexistence under most conditions.

**Environmental controls on nitrification rates and the abundance of nitrifiers**
We compared the measured nitrification rates and nitrifier abundance with concurrently measured
or available environmental factors including temperature, oxygen, light, and N concentration
($NH_4^+$, $NO_2^-$, $NO_3^-$) to assess the environmental controls on nitrification and nitrifiers (Figures 28-
31). We acknowledge that nitrification rates and nitrifier abundance are regulated by multiple
environmental factors, which may not be revealed by the simple correlation analysis with
individual factors. The new database will facilitate more sophisticated future analyses.

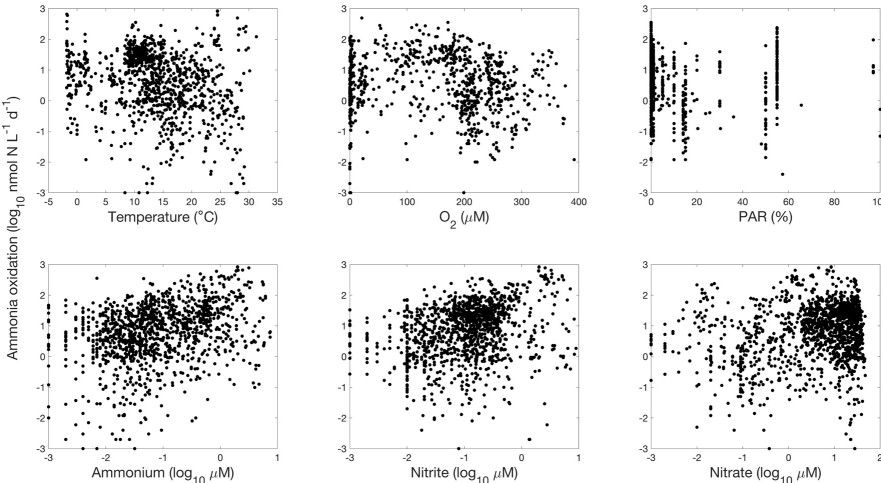


Figure 28. Relationship between ammonia oxidation rates and environmental factors.

Ammonia oxidation rates did not show a clear pattern with temperature (r=-0.22, p<0.01; Figure
28). Some of the high rates are found below 0ºC, and at around 10ºC and 25ºC. Temperature
manipulation experiments showed varying temperature sensitivity of ammonia oxidation in
different regions or among natural assemblages (Baer et al., 2014; Horak et al., 2018; Zheng et al.,
2020). The highest ammonia oxidation rates were found in the oxygen range between 100 and 200
μM (p>0.01). But ammonia oxidation has also been detected in low oxygen waters (e.g., <10 μM)
in the oxygen minimum zones (Bristow et al., 2016a; Peng et al., 2015), reflecting the high affinity
of ammonia oxidizers for oxygen. Oxygen production by ammonia-oxidizing archaea may support
their presence and activity in the oxygen minimum zones (Kraft et al., 2022). Ammonia oxidation
generally decreases at relatively high light intensity (PAR% relative to surface PAR) due to light
inhibition and substrate competition with phytoplankton (but the negative slope is not significant,
p > 0.01). Nevertheless, high ammonia oxidation rates have been measured in the euphotic zone
at 55% PAR in the Atlantic Ocean (Clark et al., 2008; Clark et al., unpublished). Ammonia
oxidation increases with N nutrient concentration (p<0.01). $NH_4^+$ is the substrate while $NO_2^-$ is the
product of ammonia oxidation. The Michaelis-Menten-like kinetics of ammonia oxidation rate
have been observed in various ocean regions (Frey et al., 2022; Newell et al., 2013; Horak et al.,
2013; Xu et al., 2019; Zhang et al., 2020; Mdutyana et al., 2022a and b). High concentrations of



$NH_4^+$ and $NO_2^-$ likely reflect intense recycling of organic matter and remineralization. The
presence of high $NO_3^-$ concentration may relieve the competition between ammonia oxidizers and
phytoplankton for $NH_4^+$, therefore leading to high ammonia oxidation rates (Wan et al., 2018). In
addition, recent studies have shown that AOA have a high requirement for iron and copper, which
may affect the distribution of nitrification in the ocean (Shafiee et al., 2019; Shafiee et al., 2021).

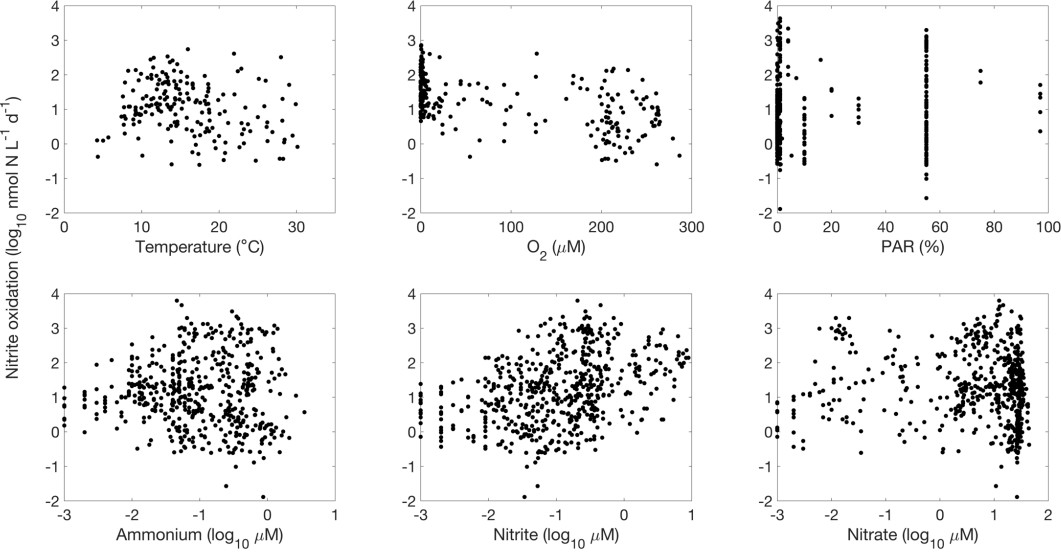


Figure 29. Relationships between nitrite oxidation rates and environmental factors.

High nitrite oxidation rates are found between 10ºC and 20ºC (Figure 28). Surprisingly, some of
the highest nitrite oxidation rates were measured in the oxygen minimum zones even with oxygen
levels below detection limits (Ward et al., 1989; Sun et al., 2017; Sun et al., 2021). Nitrite oxidation
in anoxic waters has been observed to be inhibited (Sun et al., 2017) or stimulated (Bristow et al.,
2016a) by the addition of oxygen. The mechanisms for apparently anaerobic nitrite oxidation
remain to be determined (Sun et al., 2023). Similar to ammonia oxidation, nitrite oxidation is often
reported to be inhibited by high light levels, but the relationship is not statistically significant
across the database (p>0.01; Figure 29) partly due to the presence of high nitrite oxidation rates in
the euphotic zone (e.g., Clark et al., 2016). High nitrite oxidation rates are often observed in regions
with high $NO_2^-$ concentration (r=0.23, p<0.01). For example, the highest nitrite oxidation rates
were observed at $NO_2^-$ concentrations near 0.5 µM (Figure 29).

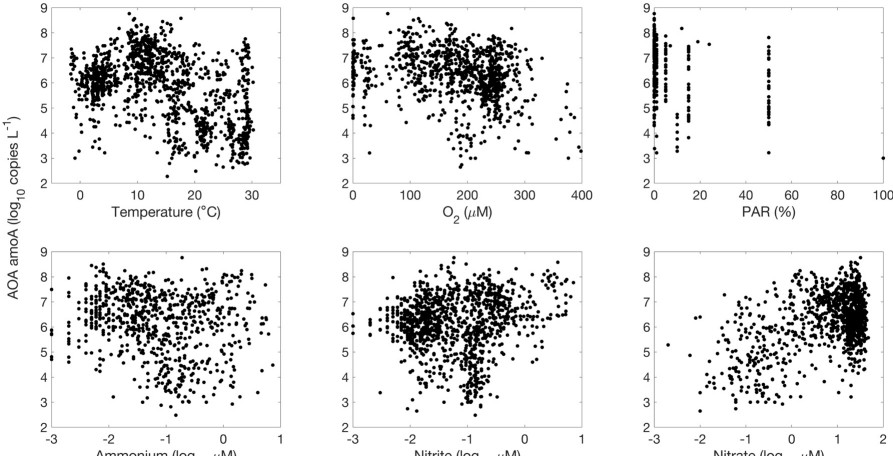


Figure 30. Relationship between archaeal *amoA* gene abundance and environmental factors.

We use *amoA* gene abundance to represent the abundance of ammonia oxidizers with the caveat
that the number of gene copies may not equal the cell numbers. Ammonia oxidizers are adapted to
a wide range of environmental conditions (Figure 30). Their abundance reaches a maximum at
around 10ºC. Ammonia oxidizers are also present in low oxygen waters and the euphotic zone
with slightly lower abundance. Interestingly, ammonia oxidizers show relatively constant
abundance across the $NH_4^+$ concentration gradient while ammonia oxidation rates are low under
low $NH_4^+$ concentration (e.g., <0.01 μM).



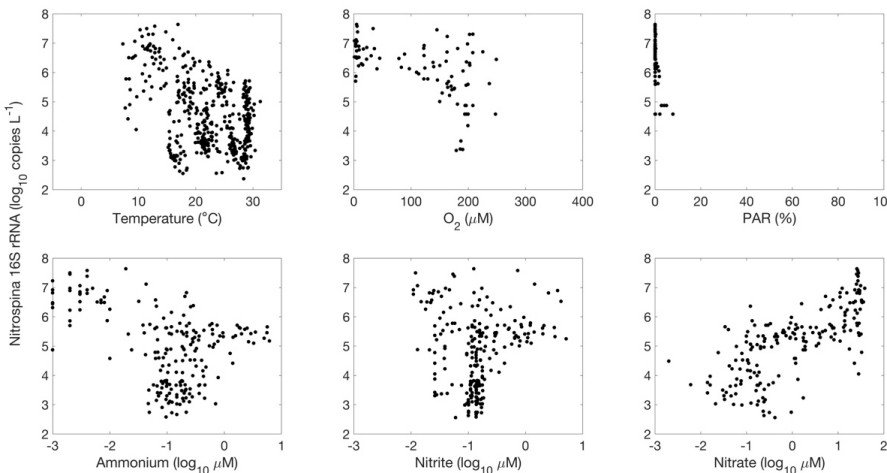


Figure 31. Relationships between *Nitrospina* 16S rRNA gene abundance and environmental factors.


It is difficult to evaluate the relationship between nitrite oxidizers and environmental factors due to the limited number of observations (Figure 31). Nevertheless, one interesting pattern is the presence of high *Nitrospina* abundance in oxygen depleted waters. The nitrite oxidizers present in the oxygen depleted waters are distinct from those found in oxygenated waters or currently cultivated strains (Sun et al., 2019; Sun et al., 2021).


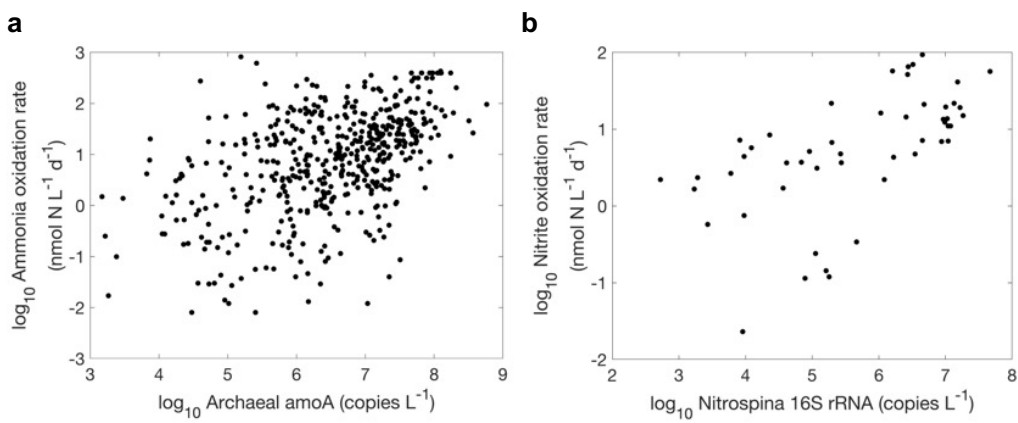




Figure 32. Relationships between nitrifier abundance and nitrification rate. (a) ammonia oxidation
vs AOA *amoA* gene abundance (y=0.46*x-2.08, r=0.48, p<0.01); (b) nitrite oxidation vs
*Nitrospina* 16S rRNA gene abundance (y=0.35*x-1.27, r=0.59, p<0.01).

There is a positive correlation between AOA *amoA* gene abundance and ammonia oxidation rates
(r=0.44, p<0.01), and between *Nitrospina* 16S rRNA abundance and nitrite oxidation rate (r=0.58,
p<0.01) (Figure 32) even though the correlation is weak. This lack of a strong relationship has also
been found in regional studies (Tolar et al., 2020), which may be caused by the perturbation of the
microbial community during rate measurement incubations. Furthermore, the addition of nitrogen
substrate during rate measurement incubations may stimulate the growth of nitrifiers and the
subsequently measured nitrification rate. Overall, using functional gene abundances to predict their
functional activity needs to be conducted with caution since the presence of genes only reflects the
functional potentials.

**Applications of the database and suggestions for future research**
This database will be useful for a broad scientific community who are interested in marine
biogeochemistry and marine microbial ecology. Potential applications include but are not limited
to: 1) Compare future measurements of nitrification rates or nitrifier abundance in a specific region
to previous measurements conducted in the same regions, and contextualize new measurements at
the global scale. 2) Meta-analysis of environmental controls on the distribution of nitrification and
nitrifier abundance at regional and global scales. The simple correlation analyses provided here
only considered individual environmental drivers separately while other drivers are changing
simultaneously. Analysis with environmental assemblages could complement laboratory culture
experiments. 3) Help to validate and improve model parametrization of nitrification and nitrifiers.
For example, ammonia oxidation has been modeled as a function of substrate concentration and
specific ammonia oxidation rate (Yool et al., 2007). However, nitrification has been found to be
regulated by temperature, oxygen, light and many other environmental factors that are not
considered in that model. A better representation of nitrification in ocean biogeochemical models
could help to constrain the estimates of marine new production, $N_2O$ production and many other
key processes. 4) Guide future nitrification studies, e.g., conducting measurements in poorly
sampled regions or seasons.




Based on the historical measurements of nitrification and nitrifiers, we provide recommendations
for future research below.

1. Method standardization is useful for comparison among studies. Nitrification has been mostly
measured by $^{15}N$ substrate tracer addition and product dilution methods. The $^{15}N$ product dilution
method tends to yield higher rates relative to the $^{15}N$ substrate tracer addition method (Figure 1).
This is perhaps to be expected since the $^{15}N$ product dilution method measures all the $NO_2^-$
production pathways including ammonia oxidation (and organic N oxidation) and nitrate reduction
instead of only $NO_2^-$ production from ammonia oxidation as is measured by the $^{15}NH_4^+$ tracer
addition method. Comparison between different methods should be conducted to resolve the
difference or even correct some of the previous measurements.

Additionally, the amount of tracer added should be recorded and reported because the increased
substrate concentration may enhance nitrification rate. Therefore, the measured rates should be
interpreted as potential rates rather than in-situ rates when the amount of tracer addition is large
compared to the ambient substrate concentration. If possible, substrate kinetic experiments should
be conducted for in-situ rate calibration (e.g., Wan et al., 2018; Mdutyana et al., 2022a and b).

The measured product of ammonia oxidation should also be reported (e.g., either only $NO_2^-$ or
$NO_2^- + NO_3^-$). When ambient $NO_2^-$ concentration is low, the $^{15}NO_2^-$ produced from $^{15}NH_4^+$ tracer
may be further oxidized to $^{15}NO_3^-$. Thus, nitrification may be underestimated if only $NO_2^-$ is
measured. Alternatively, $NO_2^-$ carrier may be added into the incubation to 'trap' the produced
$^{15}NO_2^-$. In addition to only measuring ammonia oxidation, more observations of nitrite oxidation
are desirable to evaluate mechanisms controlling the coupling or decoupling of the two steps of
nitrification.

Furthermore, measurements with at least three time points are preferred during the incubation time
courses in order to examine whether the rate has changed during the incubation period. Depending
on the incubation period, nitrification rates are reported as either nmol N $L^{-1}$ $d^{-1}$ or nmol N $L^{-1}$ $h^{-1}$.
A conversion factor (e.g., 12 or 24 hours) is required to obtain the same unit. The choice of the



conversion factor may be critical if there is a diel cycle of nitrification rate, e.g., in the euphotic
zone where light/solar radiation varies diurnally (Wan et al., 2021). Therefore, incubation
conducted under both light and dark conditions may be preferable to obtain the daily nitrification
rates. The detection limit of rate measurements should also be estimated and reported (Santoro et
al., 2013) instead of presenting rates that are below detection limit as zero.

For in-situ rate measurements, incubations should mimic the in-situ environmental conditions as
closely as possible, e.g., using light filters to simulate in-situ light/solar radiation intensity and
quality; using a temperature-controlled incubator to simulate the in-situ temperature. Particularly
for samples collected in the oxygen minimum zones, oxygen concentration in the incubation
containers should be measured or monitored throughout the incubation because oxygen
contamination is common during the sampling process (Garcia-Robledo et al., 2021). Samples
collected from the anoxic layer of the oxygen minimum zones need to be purged with helium or
nitrogen gas to remove any oxygen contamination before incubation.

2. Various primers have been designed to target ammonia oxidizers. However, current primers
miss the *Nitrosopumilus*-like *amoA* (Tolar et al., 2013; Hiraoka et al., preprint) and such group
accounts for a large fraction of the AOA based on 16S rRNA sequencing (Tolar et al., 2020). New
primers or techniques need to be developed to cover the diverse groups of ammonia oxidizers. In
addition, the quantification of nitrite oxidizers is limited. Developing primers for *nxr* genes may
be useful to untangle the relative contribution of different nitrite oxidizers particularly for the
unique ones found in the oxygen minimum zones. The report of qPCR assay should follow the
MIQE guidelines (Bustin et al., 2009) including the amplification conditions, amplification
efficiency, detection limit and other parameters. Alternatively, the abundance of nitrifiers may be
determined with quantitative metagenomics (Lin et al., 2019; Satinsky et al., 2013). In comparison
to the gene presence, gene expression and protein synthesis may be better linked to the activity of
nitrifiers (Tolar et al., 2016; Frey et al., 2022; Saito et al., 2020), deserving more observations.

3. Future observations should target regions that have been poorly sampled and regions that are
experiencing or expected to experience dramatic changes. For example, the Indian Ocean has the
fewest number of observations of nitrification and nitrifiers. With regards to change, oxygen



minimum zones are projected to change under future climate (Breitburg et al., 2018; Busecke et
al., 2022). Polar oceans (Arctic Ocean and Southern Ocean) are experiencing warming, ice melt
(which affects light/solar radiation availability) and ocean acidification (Meredith et al., 2019).
Upward nutrient supply into the subtropical gyres may be affected due to enhanced stratification
(Li et al., 2020). How nitrification will respond to these changes deserves further exploration.

Time-series studies, observations across a large-scale transect, and observations at a mesoscale or
submesoscale would be desirable for investigating the temporal and spatial variation of
nitrification rates and nitrifier abundances. When possible, both nitrification rates and nitrifier
abundance should be measured at the same locations. While this approach incurs logistical and
financial complications in requiring collaborations among laboratories with different expertise, the
benefit to comprehensive process description is manifold.

4. Incubation conditions (mentioned in point 2) and ambient environmental conditions associated
with rate measurements or gene quantification should be recorded and reported (e.g., temperature,
light, substrate concentration, oxygen). This information would be helpful for comparison among
different studies and future meta-analyses of environmental controls on nitrification and nitrifiers.
For example, light/solar radiation should be reported as both absolute light/solar radiation intensity
and relative light/solar radiation intensity to the surface ocean. Analysis of trace metals like iron
and copper concentration will be useful to assess their impact on nitrification. Standard notation
should be used to denote measurements below detection limit or measurements not conducted,
e.g., BDL for below detection limit, NM for not measured, empty/NA for data not available. A
data compilation template is provided for anyone who is interested in contributing to the database
with new datasets or datasets currently not included in the database. We encourage the scientific
community to contact us with suggestions to improve the database and to contribute to the database
with new datasets or datasets currently not included in the database.

## Data availability

Data described in this manuscript can be accessed at Zenodo repository under data doi:
https://doi.org/10.5281/zenodo.7942922 (Tang et al., 2023).




## Conclusions

We present a newly compiled database of nitrification rate and nitrifier abundance measurements
in the global ocean. This database sheds light on the spatial and temporal pattern of nitrification
and nitrifiers even though the spatial and temporal coverages remain limited. In recent years,
observations have expanded into oxygen minimum zones and polar oceans while the Indian Ocean
and Pacific Basin remain poorly sampled, especially with regard to nitrite oxidation and nitrite
oxidizers. This database can be applied to assess the environmental controls on nitrification at
regional and global scales, to validate and develop biogeochemical models, to guide future
observational efforts, and to better constrain the distribution of nitrification and assess its impact
on the marine ecosystem and climate. This database has been deposited into the Zenodo repository
and can be updated with new datasets.

## Author contributions

Weiyi Tang and Bess Ward designed the study with input from Fabien Paulot and Charles Stock.
Weiyi Tang compiled the database with data contribution from coauthors, and Weiyi Tang
analyzed the database. Weiyi Tang and Bess Ward wrote the manuscript with contribution from
coauthors.

## Competing interests

No competing interest is declared.

## Acknowledgements

We want to thank all the authors who have kindly shared data for this community effort. Weiyi
Tang, Bess Ward, Fabien Paulot, and Charles Stock are funded by Cooperative Institute for
Modeling the Earth System (CIMES). Gerhard J. Herndl is funded by the Austrian Science Fund
(FWF) project DEPOCA (P 35587-B).



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
