# Peer review of "Database of nitrification and nitrifiers in the global ocean"

_Earth System Science Data, 2023_

## Author Comment (AC1)

Reply to reviewers' comments on **Database of nitrification and nitrifiers in the global ocean**

We thank the reviewers for their constructive comments that helped to improve our manuscript. The detailed response to each comment is shown in blue below.

Reviewer 1:
Overall, this manuscript presents a well-compiled nitrification and nitrifiers database in the global ocean. Statistical analyses of the distribution patterns of nitrification rates and nitrifier abundance were provided, along with a brief summary of the influence of different environmental factors, such as temperature, pH, light, N concentration etc. This database has the potential to make significant contributions to the advancement of ecological and biogeochemical models.

**Strengths:**
1. The data and methods presented in the manuscript are new and have the potential to be useful in the future. The authors have compiled a database of nitrification rates and nitrifier abundance in the global ocean from published literature and unpublished datasets, which can be used to constrain the distribution of marine nitrification, evaluate and improve biogeochemical models of nitrification, and quantify the impact of nitrification on ecosystem functions like marine productivity and $N_2O$ production.
2. The methods and materials are described in detail, allowing other researchers to replicate the study or use the database for their own research.
3. The references and citations to other data sets or articles appear to be appropriate and complete.
4. The article itself is appropriate to support the publication of a data set, as it provides a description of the data collection, processing, and analysis methods.
5. The data set itself appears to be of high quality, as it is accessible, and well-documented. The data set is usable in its current format, and the formal metadata is appropriate.

Thus, I recommend accepting this manuscript after some major revisions based on the suggestions provided below.

We thank the reviewer for the positive comments.

**Major:**
(1) **Method.** The rate values were mainly obtained by using isotope tracers. The incubation condition, preincubation environment, and calculating methods would have a large impact on the final result. In line 204-214, the author summarized the general incubation procedure for nitrification, which is appropriate in the main text. However, during the incubation, different controlling conditions existed in different studies, especially of temperature, light (or paired light and dark to calculate daily), N concentration, and tracer concentration. These essential information (if available) should be directly present in the data table (not only in metadata) for further interpretation and analysis. In addition, calculating equations should also be presented in the data file instead of only in the reference.

We agree with the reviewer that incubation conditions will affect the rate measurements. Due to the limitation in space in Table 1, we provided the incubation condition in the database file. In

the main text, we added "Detailed incubation conditions for each study are presented in the database file". The equations used to calculate nitrification rates are now added in the manuscript.

**(2) Data presentation.** Some trends are "buried" in the scatter plot. This manuscript is actually too long for a dataset paper (32 figures and 66 pages). An overly lengthy analysis may turn off some interested readers. I have some suggestions:

1. **a)** Some "below detection limit" data should be also presented and marked in proper ways (like marked proportion).

We have now specified the number of observations that are below detection limit or noted as zero in the original publication: "After removing measurements of zero and below detection limit (277, 132, 51, 240, 6 and 11 observations for ammonia oxidation, nitrite oxidation, AOA *amoA*, AOB *amoA*, 16S rRNA of *Thaumarchaeota* and *Nitrospina*), nitrification rates and nitrifier abundances were log10 transformed before further analysis".

Data below detection limit are shown in the black circles on the map of spatial distribution of nitrification rates and nitrifiers abundance, e.g., Figures 7, 12, 22 and 26.

2. **b)** This manuscript mainly analyzes the scattered points of nitrification data, and I did not see the analysis of water column integration, considering that each nitrification profile must go through the whole numerical "cycle" from 0 to the maximum value (Figure. 13). The analysis of scatter data can lead to a shot-gun situation. I suggest that the currently available profile data should be analyzed by water column integral processing under a certain standard, which may achieve better results. Alternatively, please consider merging the scatter plot and box plot in the text into a single one (such as Figures 4 and 5), where the scatter plot can serve as the background for the box plot. This will enhance the clarity and conciseness of the graphical presentation in the article. Or try to remove some into the appendix or supplementary.

It's a great suggestion to look at the nitrification rate integrated over the water column. However, the integrated nitrification is difficult to estimate due to the large variation in the sampling depth for each profile across studies. For example, some studies only measure nitrification in the top 200 meters while others extend to below 2000 m. We acknowledge that the vertical distribution of nitrification has been fitted with a Martin Curve or exponential decay (e.g., Ward and Zafiriou. 1988; Newell et al., 2011). Analysis of depth-integrated nitrification rates deserves further exploration but is beyond the scope of this data description paper and would only exacerbate the length problem.

Scatter plots show a detailed distribution of all the data points, which would supplement the box plot (see also the comment from Reviewer 3). We decided to keep the scatter plot and added jitter to the plot when applicable (Figures 5, 10, 19, 24).

3. **c)** Scatter plots should be color-classified by different regions to make the image more orderly

Based on the reviewer's suggestion, we color-coded the points by the observation regions in the scatter plots (Atlantic Ocean, Pacific Ocean, Indian Ocean, Arctic Ocean and Southern Ocean). We found the color-coded points to be useful in the figures showing the environmental controls

on nitrification and nitrifiers (Figures 28-31). Below is an example for the correlation between environmental factors and ammonia oxidation rates observed at different ocean basins.

[Figure]

4.  **d)** Subgraph numbers should be added in all combined graphs. This will help readers easily refer to specific panels in the graph and enhance clarity in the presentation of the data.
We have now added subplot numbers to each graph.

5.  **e)** In Figures 16, 17, 21, and 22, it is unclear whether the amoA archaeal gene abundance corresponds only to the total AOA in the database. Additionally, certain categories such as "Shallow clade AOA", "WCA", "Deep clade AOA", "low-ammonia group AOA", and "high-ammonia group AOA" in Figure 21 are not specifically marked in the database. I suggest that all labels in figures to be consistent with the database to ensure direct access to the corresponding information in the database.
Thanks to the reviewer's comment, we clarify the archaeal amoA gene abundance in the captions of Figure 16, 17, 21, and 22, which is the total number of amoA gene or the sum of water column group A and B. In addition, we now specify the AOA group in the database based on the reviewer's suggestion: shallow clade AOA include both WCA and low-ammonia group AOA while deep clade AOA include WCB and low-ammonia group AOA.

"Within the measurements of AOA *amoA* abundance, 1204 and 1101 measurements were separately conducted for water column ecotype A (WCA) *amoA* and water column ecotype B (WCB) *amoA*. Thus, the total *amoA* gene abundance was calculated by summing the abundance of WCA and WCB when available".

"Since total AOA *amoA* genes have the largest number of observations and better represent ammonia oxidation capability, we will use it to show the spatial and vertical distribution of ammonia oxidizer abundance".

**Minor:**

1. Please note that the submission guidelines: Genus and species names are italic; high-order taxonomic ranks are roman. Thus, all graph Labels about the gene names and microbial names should be using italics, such as, "amoA" and "Nitrispina" in Figures 15-27 should be changed to "*amoA*" and "*Nitrispina*".

We have updated the format of genes' or organisms' names in all figures to meet the submission guidelines.

2. Line 77 and Line 80: There is inconsistency in the usage of nitrifiers or nitrifiers'. It is recommended to be consistent throughout the text.

We used <nitrifiers'> as the possessive and <nitrifiers> as the plural, but agree that is somewhat awkward. So we have changed the wording to avoid the possessive and use nitrifier or nitrifiers for appropriate elsewhere.

3. Line 143 Table 1 and 147 Table 2, Should be changed to a three-line table format. These modifications will enhance visual clarity.

Tables 1-3 have now been reformatted to three-line table.

4. Line 628 $p > 0.01$ and Line 622 $p>0.01$, To ensure consistency in the formatting of symbols and text, it is recommended to include spaces between symbols and text throughout the entire text.

Spaces are now added among symbols and text.

5. Table 3: Water Column ecotype A (WCB) should be Water Column ecotype B (WCB).

Thank the reviewer for catching the error. It has been modified to "Water Column ecotype B".

6. Make a consistent mark in the datasheet. For example, I see "light", "in situ", and "yes" in the "in-situ light simulation" column of "rate metadata". Make sure you don't confuse your readers.

We have updated the database spreadsheet to make the marks consistent across different environmental conditions. For example, if "in-situ light simulation" was conducted in the original paper, it is now marked as "Yes".

**Reviewer 2**
**Summary**
The authors have compiled an impressive database of hundreds to thousands of environmental rate measurements on both ammonia and nitrite oxidation in the ocean, together with reports of quantitative measurements of the abundances of the major microbial groups presumed to be carrying out these processes.

The synthesis of such a dataset is exciting, necessary and offers the community an important tool for capturing the big picture of this important process. Despite the wide range of techniques and methodologies used, the synthesis provides good assessment of major patterns across the regions of the ocean where studies have been focused. In this sense, this synthesis also provides an excellent picture of where the biggest gaps are – and thereby offers a useful roadmap for the next set of studies that need to be conducted to close such gaps. Although there were no major 'discoveries' that dropped out of such a synthesis – this will prove to be a valuable contribution for sure.

I appreciated the attention given to the limitations in methodologies and approaches as well as the recommendations listed for how to optimize future work in this field. This will make a valuable contribution to the literature. I have a few recommendations below for areas that might be clarified or modestly expanded on, as well as a few typos and editorial comments throughout that I hope are useful.

We thank the reviewer for the positive comments. Modifications have been made to the manuscript based on the suggestions.

**Major comments:**
1. My one primary observation that I feel was glossed over by the authors is in the attribution of nitrification rate regulation by light limitation. Both Figure 6c and Figure 11c clearly seem to indicate that rates are actually highest in that upper 10m of the sunlit ocean. For nitrite oxidation – it seems this was perhaps explained by the high rates reported by Clark *et al*., for the UK shelves, subpolar N Atlantic and Mauritaneian upwelling system. It is unclear whether these are the same studies that used the $^{15}$N dilution method which appears to have a systematic divergence from nitrite oxidation rates measured through $^{15}NO_2^-$ addition (Lines 193-195). In Figure 13b, are these left out? If so – this would be good to clarify (i.e., clarify why Fig 13b appears to be a bit different from 11c)? For ammonia oxidation is there a similar explanation for why the highest rates of ammonia oxidation in the whisker-bar plot in Fig 6c 'seem' to be absent from the Figure 13a? All in all, I was just looking for a bit more clarity on this surface ocean nitrification dynamic from this database – and felt a little confused by some of how the data were represented.

We thank the reviewer for the positive comments. It's a very good observation by the reviewer that the light regulation on nitrification is ambiguous at the global scale. This uncertainty may be caused by the covarying environmental drivers like substrate or temperature along with light, by the methodological difference across studies, and other unidentified factors. For example, as the

reviewer pointed out, some of the high nitrification rates in surface ocean were measured using isotope dilution method (e.g., Clark et al., 2014, 2016 and unpublished; Cavagna et al., 2015).

Figures 6 and 11 showed all the available data. However, Figure 13 presents the vertical profiles of nitrification rates (i.e., each line represents a vertical profile of nitrification rate with at least 5 depths measured at one location). Therefore, some studies that only measured the surface ocean (e.g., euphotic zone) with less than 5 depth data points are not included in Figure 13 (e.g., Clark et al., 2014, 2016 and unpublished). We have added clarification in the text and figure captions.

2.  Similarly, despite text that suggests that the distribution of AMO rates and AOA as well as nitrite oxidation rates and NOB were qualitatively similar, respectively, - I was struck by the distinction between Figures 20c and 6c, and 25c and 11c. In comparing these pairs of figures, rates appear to be quite elevated with respect to the clearly lower abundances of organisms at these shallowest depths above 60m. Some clarification on this dynamic might be useful for understanding nitrification in the surface ocean.

Not all the rate and abundance measurement are conducted at the same time and same place. Particularly, many studies that reported high nitrification rates in the surface ocean did not quantify the abundance of nitrifiers.

In addition, the difference in vertical profiles of nitrification rate and nitrifier abundance may also reflect differences in the cell-specific nitrification rates. For example, cell-specific ammonia oxidation rate (assuming 1 *amoA* gene copy per cell) varied substantially with depth and tends to be higher in the surface 200 m with large variation, as does cell-specific nitrite oxidation rate (see figure below). The vertical variation of cell-specific nitrification rates remains to be evaluated but is beyond the scope of this study.

We have added clarification in the text: "We noticed that *amoA* abundance and ammonia oxidation rates appear to have different depth distributions, particularly for the top 200 m (Figure 6c and Figure 20c): *amoA* abundance in 0-10 m layer is lower than in 100-200 m layer while ammonia oxidation rates in 0-10 m layer are comparable to the rates observed in 100-200 m layer. These distributions may suggest depth differences in cell-specific activity which might be interesting for future investigation".

[Figure]

3. Given the large uncertainty associated with the stoichiometric conversion of C to biomass during nitrification – it is not clear to me whether rates derived from the $^{14}HCO_3^-$ incorporation experiments are as justifiably included in this database. Personally, I would recommend leaving these out of the database to help keep the methodologically sourced 'noise' down.

We would like to keep the data derived from $^{14}HCO_3^-$ incorporation experiments in order to show the comprehensive measurements by different methods. We laid out the method limitation and the readers can decide which data to use or compare. Presenting the range of data should help users pick the best method.

4. This might be a useful forum for a couple paragraphs on the nuances underlying the calculations of these rates – as alluded to on L249-253. In particular, even just spelling out some of the basic sets of equations could be useful for helping guide readers through what aspects might underlie variability among the different studies.

We have provided the different equations used to calculate nitrification rates and also provided further references for readers if they are interested in detailed derivation of equations.

**Minor comments:**

Many of the figures did not clearly reproduce (blurry, fuzzy) in the PDF version – somewhat limiting the utility of their color coding and geographic distributions (e.g., Fig 2a; Figure 7).

High-resolution figures will be provided for the final version of the manuscript for publication.

L70: group
Modified.

L93: uptake
Modified.

The headers for Table 1 are unclear. It seems likely there was a formatting error.

The headers of Table 1 and 2 are reformatted. They have also been changed to three-line table based on the comment from the other reviewer.

L321: Wording: The quantification of nitrifier abundance starts to accumulate after 2002.

Modified.

L323: originates

Modified.

L385: light limitation of nitrifiers

Modified.

L390: missing period

A period is added at the end of the sentence.

Figure 15 – please explain what is implied by the lighter shaded nitrite oxidation rates (I assume that these refer to the $^{15}$N dilution method estimates which may be biased?).

The orange bars (both light and dark) represent the observation of nitrite oxidation. The dark orange color is due to the overlap with blue bars - ammonia oxidation. We have now clarified in the figure caption.

Figure 16 – please explain the lighter shaded AOB bars.

The lighter and darker orange bars are both AOB. The darker color is caused by the overlap with the blue bars – AOA. This is now explained in the figure caption.

L625: I'm not sure that Figure 28c (PAR) supports the statement that ammonia oxidation rates decrease at high light intensity.

We have now modified the statement about the impact of light on nitrification: "Although light manipulation experiments have shown clear light inhibition of nitrification rate at specific locations (e.g., Xu et al., 2019; Shiozaki et al., 2019), the relationship between nitrification and light intensity is ambiguous at the global scale, which may be related the compounding factors on nitrification. For example, the covarying ammonium availability would complicate the impact of change in light intensity".

Figure 30 and 31: The panels comparing AOA or Nitrospina abundances vs NO$_3^-$ are not discussed.

We have added the discussion comparing AOA or *Nitrospina* abundance with NO$_3^-$ concentration: "A large portion of the *amoA* observations were conducted in the deep ocean where nitrate concentration was above 10 μM. Some of the highest *amoA* abundance were found in these NO$_3^-$ enriched waters". "Similar to *amoA* abundance, *Nitrospina* 16S rRNA gene abundance also increased with NO$_3^-$ concentration".

Lines 678 – 682: There are some discrepancies in here with respect to correlation coefficients (r values).

Thank the reviewer for catching the discrepancies in the correlation coefficients. We have updated the values.

L755: … and this group accounts…
Modified.

**Reviewer 3**

in their manuscript entitled "Database of nitrification and nitrifiers in the ocean", Tang and colleagues have compiled a large set of previously published and several unpublished datasets of ammonia and nitrite oxidation rate measurements, as well as qPCR datasets on the abundance of the ammonia and nitrite oxidizing organisms. The authors briefly review the implications of the different methods used to generate the data, and use their compiled data to investigate trends in the activity and abundance of niitrifying organisms across ocean regions, depth, latitude, and seasons. Correlations between abundance of nitrite and ammonia oxidizers, and between organism abundance and activity are also shown. Finally, recommendations are made for future experimental design and target regions, to maximize the value that can be gained from comparative and synthesis studies.

Overall, this data synthesis is a valuable effort to show the state of our measurements of nitrification in the global ocean, and to guide future studies.

We thank the reviewer for the positive comments and we modified the manuscript based on the comments as described below.

I have some recommendations that I hope can be addressed before final publication:

At the moment, it isn't immediately clear from the database which studies have coupled measurements of ammonia and nitrite oxidation, and which studies also include quantification of either ammonia or nitrite oxidizers. Table 2 and 3 cover in the manuscript cover this to a certain extent, but I think it would be beneficial to add this information to the database as well. This could be done in the form of a few extra columns in the "volumetric ammonia oxidation", "volumetric nitrite oxidation", "ammonia oxidizer", and "nitrite oxidizer" tabs, indicating whether coupled measurements are available or not (Y/N).

This is a good suggestion. In Table 2, we have added two columns – "ammonia oxidizer" and "nitrite oxidizer" to show if both rate measurements and qPCR data are available. Similarly, in Table 3, we have added two columns – "ammonia oxidization" and "nitrite oxidation".

for the outlier removal, the authors use Chauvenet's criterion, but don't show whether the data is normal or lognormal distributed. As far as I can tell from the text, the only excluded values where high and low outliers in the ammonia oxidizer qPCR quantification. Instead of listing the examples, I would recoomend listing how many observations were removed from each dataset, and for what reason (outlier or zero/below detection).

Nitrification rates and nitrifier abundance showed normal distributions after log-transform (see Figures 15, 16 and 27). Following the reviewer's suggestion, we have specified the number of observations removed as outliers or below detection limit.

"After removing measurements of zero and below detection limit (277, 132, 51, 240, 6 and 11 observations for ammonia oxidation, nitrite oxidation, AOA *amoA*, AOB *amoA*, 16S rRNA of *Thaumarchaeota* and *Nitrospina*), nitrification rates and nitrifier abundances were log10

transformed before further analysis. Nitrification rates and nitrifier abundances reported at 0 or below detection limit are noted separately in the database and following analysis."

"Some outliers were identified by Chauvenet's criterion for ammonia oxidizers (1 for AOB *amoA* and 1 for 16S rRNA of *Thaumarchaeota*)".

Please check the colors in figure 1 and 2 for colorblind friendliness

These color schemes in Figures 1 and 2 are colorblind friendly.

In figure 3a, it seems a 5th category (Thaumarcheaota 16S + archaeal amoA + bacterial amoA) seems to have been omitted, even though there seem to be several studies in the database that characterized all 3

We have added another category called "archaeal and bacterial *amoA*, and 16S" (see figure below).

[Figure]

In figure 5, 10, 19, and 24, please specify in the legend what the x-axis numbers refer to, or label the months by a 1 letter or 3 letter abbreviation. I assume 1 corresponds to january, but it is not stated anywhere.

In the figure caption, we added: "monthly variation (1-12: January to December)".

for those same figures, I understand that the strip plots have been added here for consistency

with all other figures, and I generally like the combination of scatter and box plot as used in the manuscript. However for the data by month both the strip plot and the box plots are categorical. if showing the strip plot, consider adding jitter to the data so the distribution is more clear.

Following the reviewer's comment, we have added jitters to the categorical plots (i.e., monthly plots). See below monthly variation of ammonia oxidation rates as an example.

[Figure]

Most figures that include depth information use an axis break at 500m, and show the deepest measurements at a different axis scale, please mention this in the legend where appropriate. Figure 8 and 14 do not use an axis break, instead showing the shallower samples (up to 500m depth) in a seperate panel. Either is fine, but I would suggest using one consistently throughout the manuscript.

We have now added description of the axis breaks in figure captions.

I noticed figure 21 is not mentioned in the text, please check throughout that all figures are discussed in the text.

We have now referred Figure 21 in the following sentence: "Water column group A dominates the upper 200 meter while water column group B is more abundant in the mesopelagic and

bathypelagic deep ocean below 500 m (Figure 21)".

line 267: the phrase "partially encoded" struck me as odd, perhaps consider changing to "which is a multisubunit enzyme partially encoded by the amoA gene"

We have applied the suggestion to the sentence.

line 553-557: this paragraph seems to suggest that the lowest values are driven by geography and nutrient limitation, but it looks like depth plays a much bigger role overall, if the discussion of the lowest values is ment to refer primarily to the shallow depths, please explicitly state that in the text.

We agree with the reviewer that *amoA* abundance changed substantially with depth (e.g., in Figure 20-22). There is a spatial variation in *amoA* abundance in different vertical layers. For example, eastern tropical South Pacific had higher *amoA* abundance that in the western Pacific at both shallow and deep depths, which may be related to the difference in their primary productivity and subsequent remineralization. However, the exact mechanisms remained to be explored in future analyses.

line 629 (and likely elsewhere): perhaps it is better to refer to previously unpublished data included in this work as "this study", or "previously unpublished"?

We will leave the editorial team to decide how to cite the "previously unpublished" data.

line 712-719: This paragraph reads as if the authors are going to recommend using the tracer addition methods over the product dilution methods, but then stop short and state a comparison is necessary. Most studies incorporated in the database seem to have been conducted with tracer addition methods, so is there a specific reason the authors would not want to recommend using these going forward?

We want to clarify that both tracer addition and product dilution method are useful to measure nitrification rates. We are not recommending using the tracer addition methods over the product dilution method but emphasize the differences in what the two methods are measuring. Future studies should note the method difference especially when comparing the results from different methods. We recommend a comparison between these two methods conducted at the same time and location, which could be useful to interpret the previously measured rate differences.

line 792-793: the authors mention a data compilation template is available, but I could not find it. Perhaps add a link to the template under data availability

We have now added the data compilation template to the Zenodo repository in the link below: https://doi.org/10.5281/zenodo.8355912.

References:

Cavagna, A. J., Fripiat, F., Elskens, M., Mangion, P., Chirurgien, L., Closset, I., Lasbleiz, M., Florez-Leiva, L., Cardinal, D., Leblanc, K., Fernandez, C., Lefèvre, D., Oriol, L., Blain, S., Quéguiner, B., and Dehairs, F.: Production regime and associated N cycling in the vicinity of Kerguelen Island, Southern Ocean, Biogeosciences, 12, 6515-6528, 10.5194/bg-12-6515-2015, 2015.

Clark, D. R., Widdicombe, C. E., Rees, A. P., and Woodward, E. M. S.: The significance of nitrogen regeneration for new production within a filament of the Mauritanian upwelling system, Biogeosciences, 13, 2873-2888, 10.5194/bg-13-2873-2016, 2016.

Clark, D. R., Brown, I. J., Rees, A. P., Somerfield, P. J., and Miller, P. I.: The influence of ocean acidification on nitrogen regeneration and nitrous oxide production in the northwest European shelf sea, Biogeosciences, 11, 4985-5005, 10.5194/bg-11-4985-2014, 2014.

Newell, S. E., Babbin, A. R., Jayakumar, A., & Ward, B. B. (2011). Ammonia oxidation rates and nitrification in the Arabian Sea. *Global Biogeochemical Cycles*, *25*(4).

Shiozaki, T., Ijichi, M., Fujiwara, A., Makabe, A., Nishino, S., Yoshikawa, C., and Harada, N.: Factors Regulating Nitrification in the Arctic Ocean: Potential Impact of Sea Ice Reduction and Ocean Acidification, Global Biogeochemical Cycles, 33, 1085-1099, 10.1029/2018gb006068, 2019.

Ward, B. B., & Zafiriou, O. C. (1988). Nitrification and nitric oxide in the oxygen minimum of the eastern tropical North Pacific. *Deep Sea Research Part A. Oceanographic Research Papers*, *35*(7), 1127-1142.

Xu, M. N., Li, X., Shi, D., Zhang, Y., Dai, M., Huang, T., Glibert, P. M., and Kao, S. J.: Coupled effect of substrate and light on assimilation and oxidation of regenerated nitrogen in the euphotic ocean, Limnology and Oceanography, 64, 1270-1283, 10.1002/lno.11114, 2019.

---

## Author Response (AR2)

Reply to the second round of reviewers' comments on **Database of nitrification and nitrifiers in the global ocean**

We thank the reviewers for their constructive comments that helped to improve our manuscript. The detailed response to each comment is shown in blue below.

Reviewer 3:
The authors have generally addressed my concerns and I am pleased to see that the readability of the data tables has been greatly improved. I think the current manuscript is in an acceptable form for ESSD.

Some minor issues:

Figure 13 Nice figure. However, given the large difference in the number of observations for the two rates, the differences in the median profiles could also be brought about by regional differences, authors can consider marking the paired profiles with different colors.
We acknowledge that the median profiles of ammonia oxidation and nitrite oxidation could be biased by the number of profiles available for ammonia oxidation (203) vs nitrite oxidation (96) and the distribution of these profiles (see the number of observations separated by ocean basins in Figure 2 (and below) and discussion in lines 456-461). In fact, the relative distributions of ammonia vs nitrite oxidation rates are similar, so regional bias may not be a factor. We now use grey lines and black lines to show non-paired and paired profiles (i.e., ammonia oxidation and nitrite oxidation were measured concurrently), respectively in the updated Figure 13 and below. The median profiles are represented in red lines.

[Figure]

Figure. Number of ammonia oxidation and nitrite oxidation in major ocean basins (AO: Atlantic Ocean; PO: Pacific Ocean; IO: Indian Ocean; Ar: Arctic Ocean; SO: Southern Ocean).

[Figure]

Figure. Vertical profiles of ammonia oxidation and nitrite oxidation rates. Grey lines are non-paired measurements. Black lines show the paired observations (concurrent measurements of ammonia oxidation and nitrite oxidation). Red lines are median rates.

Lines 573-576 Ammonia are also very low in the deep sea, and the results here (Fig. 21) seems cannot reflect the different ammonia affinity of WCA and WCB

We thank the reviewer for capturing the error: WCA AOA was mislabeled as low-ammonia group AOA while WCB AOA was mislabeled as the high-ammonia group AOA in Figure 21. WCA AOA is the high-ammonia group while WCB AOA is the low-ammonia group or likely has higher affinity for ammonia, dominating in the deep ocean. The mislabeling has been corrected in Figure 21.

Line 489, 513, 629, 724-725 The formulas here seem to be exponential because x and y are taken logarithmically. Personally, I think it might make more sense for the formula here to give the relationship under linearity.

We provided the equations in log10-transformed format because the data have roughly log-normal distributions rather than normal distributions. Linear regression could be obscured by the large anomalies in a normal scale. We compared the figures with and without log10-transformed data. For example, the comparison between ammonia oxidation and nitrite oxidation is shown below. The linear fit for the log10-transformed data is $log_{10}y = 0.53 \times log_{10}x + 0.91$ (r=0.5, p<0.01) while the linear fit for the data in normal scale is $y = 0.6 \times x + 116.8$ (r=0.2, p<0.01). Due to the better correlation in the linear regression, we decided to keep the log10-transformed format.

[Figure]

I would still recommend merging some similar figures together if possible. But the decision can be left to the editor.
We would like to keep the figures separated for better visualization.